# An ArfGAP-dependent signaling modulates synaptic plasticity via IP3-regulated calcium release from the endoplasmic reticulum

Bhagaban Mallik[1,2], Shikha Kushwaha[1], Anjali Bisht[1], Harsha MJ[1], C Andrew Frank[2]*, Vimlesh Kumar[1]*

1 Department of Biological Sciences, Indian Institute of Science Education and Research (IISER) Bhopal, Bhopal, Madhya Pradesh, India, 2 Department of Anatomy and Cell Biology, University of Iowa, Lowa City, Iowa, United States of America

* andy-frank@uiowa.edu (CAF); vimlesh@iiserb.ac.in (VK)

## Abstract

Calcium release from intracellular stores influences synaptic response timing and magnitude. Despite the critical role of inositol trisphosphate (IP3)- and ryanodine receptor (RyR)-dependent calcium release in regulating synaptic strength, the upstream signaling mechanisms that govern IP3 receptor or RyR activity remain elusive. Here, we provide evidence that the ArfGAP-containing protein Asap modulates NMJ morphogenesis and synaptic calcium homeostasis by activating IP3-mediated calcium release from the endoplasmic reticulum (ER) via the phospholipase C-beta (PLCβ) signaling pathway. Using CRISPR/Cas9-engineered *Asap* mutants and genetically encoded calcium sensors, we demonstrate that loss of *Asap* leads to elevated resting synaptic calcium, resulting in increased evoked amplitude, elevated spontaneous miniature frequency, and reduced synaptic failures under low extracellular calcium conditions. Additional pharmacological and genetic manipulations of calcium regulatory pathways further support the role of increased resting intracellular calcium in driving enhanced neurotransmission in *Asap*-deficient synapses. Consistent with the role of Asap's ArfGAP domain in NMJ morphogenesis and intracellular calcium regulation, expressing a GDP-locked form of Arf6 (Arf6DN) or knocking down *Arf6* in *Asap* mutants not only rescues Asap-associated synaptic defects but also normalizes synaptic calcium levels. Furthermore, epistatic analysis revealed that attenuation of IP3-signaling components in animals constitutively expressing Arf6CA normalized the NMJ morphological defects and synaptic functions. Together, these findings provide novel insights into the role of Asap-Arf6-PLCβ signaling in IP3-regulated calcium dynamics, sustaining both structural and functional synaptic plasticity.

**Data availability statement:** All relevant data are within the paper and its Supporting Information files.

**Funding:** This work was supported by grant number BT/PR/26071/GET/119/108/2017 to VK (Department of Biotechnology, Government of India (https://dbtindia.gov.in/) and grant number NS130108 from NIH/NINDS to CAF (https://www.ninds.nih.gov/). BM was partly supported by NIH/NINDS grant NS130108 to CAF (https://www.ninds.nih.gov/). The funders had no role in study design, data collection and analysis, decision to publish, or preparation of the manuscript.

**Competing interests:** The authors have declared that no competing interests exist.

## Author summary

Calcium signaling plays a central role in how neurons communicate and adapt, yet the upstream mechanisms that control calcium release from internal stores remain poorly understood. In this study, we identify a previously unrecognized signaling pathway that regulates active zone organization and calcium dynamics at the *Drosophila* neuromuscular junctions. We show that the ArfGAP-domain containing protein Asap acts through the small GTPase Arf6 and phospholipase C-beta (PLCβ) to stimulate inositol trisphosphate (IP3)-dependent calcium release from the endoplasmic reticulum. Loss of *Asap* disrupts calcium homeostasis, leading to elevated basal calcium levels, increased neurotransmitter release, and structural changes at synaptic terminals. Suppressing Arf6 activity or blocking IP3 signaling restores normal synaptic structure and function, highlighting a critical role for the Asap-Arf6-PLCβ-IP3 pathway in maintaining basal synaptic calcium balance, active zone integrity and synaptic stability. These findings reveal a new molecular link between small GTPases and calcium signaling mechanisms that fine-tune synaptic communication.

## Introduction

Calcium signaling is essential for neuronal function, acting as a critical mediator in processes ranging from synaptic activity to neuronal survival. In presynaptic terminals, calcium influx, typically triggered by depolarization and the subsequent activation of voltage-gated calcium channels (VGCCs), initiates synaptic vesicle (SV) exocytosis [1]. Furthermore, intracellular calcium release modulates SV release probability and influences the spatial and temporal dynamics of synaptic transmission [2–4]. Multiple studies highlight the role of G-protein-coupled receptor (GPCR) signaling in calcium regulation [5–7]. Upon ligand binding, GPCRs activate intracellular pathways mediated by heterotrimeric G-proteins. For example, GPCRs coupled to Gq proteins activate phospholipase C (PLC), which generates IP3. IP3 subsequently binds to receptors on the endoplasmic reticulum (ER), releasing calcium into the cytosol [8–10]. While it is well established that calcium release through ER channels, such as RyRs and IP$_3$ receptors (IP$_3$Rs), or via mitochondrial channels like the $Ca^{2+}/H^+$ antiporter Leucine zipper and EF-hand containing transmembrane protein 1 (Letm1) can regulate synaptic calcium homeostasis, the upstream signaling pathways that regulate these channels remain elusive [8,11,12].

One of the possible mechanisms by which GPCR/PLCβ signaling might be regulated is through small GTPases, which serve as molecular switches controlling various aspects of cellular signaling [13–15]. In support of this, studies in cultured cells have shown that transient activation of small GTPases such as RhoA may increase intracellular calcium [16]. Similarly, studies in neuroendocrine cells have implicated the role of ADP-ribosylation factor 6 (ARF6) GTPase in the regulation of calcium-dependent dense core vesicle exocytosis by regulating PIP2, a phospholipid that is

crucial for IP3 formation [17]. A direct role of Arf6 in regulating PLCβ and IP3-mediated calcium release has been shown in human acrosomal fusion [18]. ARF activity is regulated by Arf-specific GTPase-activating proteins (ArfGAPs) and Guanine nucleotide exchange factors (ArfGEFs), which facilitate GTP hydrolysis and the exchange of GTP for GDP, respectively, and may regulate local calcium dynamics through modulation of Arf activity. The ASAP subfamily proteins are multidomain ArfGAPs that contain SH3, Ankyrin repeat, BAR and ArfGAP domains. They catalyze the hydrolysis of GTP bound to Arf6 through their intrinsic GTPase-activating activity *in vitro* [19,20]. Despite a possible link between small GTPases and calcium release, whether ArfGAP or ArfGEF-dependent modulation of Arfs regulates intracellular calcium release from internal stores and contributes to presynaptic functions remains unclear.

We sought to evaluate the functions of a *Drosophila* ArfGAP containing Asap, particularly in synapse development and calcium homeostasis. We found that the CRISPR/Cas9-engineered loss-of-function mutants of *Asap* resulted in increased bouton size, reduced bouton number, and higher active zone density at the NMJ. Loss of *Asap* resulted in increased mEJP frequency and reduced synaptic failures under low extracellular $Ca^{2+}$. These physiological phenotypes due to loss of *Asap* arise from elevated basal synaptic calcium through a pathway that involves IP3-induced calcium release from the ER. Furthermore, epistatic interaction analysis revealed that Rab3, a known regulator of Brp organization at *Drosophila* NMJ, acts downstream of Arf6 to modulate Brp puncta distribution. Together with calcium imaging and electrophysiological analyses, these findings indicate that Asap regulates Arf6 activity to restrict IP3-dependent calcium release and maintain the proper structure of active zones. Thus, our study identifies a novel role of Asap in regulating resting presynaptic calcium through Arf6 and IP3-dependent pathways to control synaptic growth and sustain vesicle release probability.

## Results

### ArfGAP containing Asap functions in neurons to regulate synaptic bouton morphogenesis

A targeted, small-scale, RNAi-mediated screen of *Drosophila* BAR-domain-containing proteins indicated a possible role for Asap in synaptic morphogenesis [21]. In order to analyze the synaptic functions of Asap, we first created loss-of-function mutants of *Asap* using CRISPR/Cas9-mediated genome editing. From this approach, we obtained two viable alleles, *Asap^K23* (with 4190 bp deletion) and *Asap^B52* (with 4275 bp deletion) (Fig 1A and S1Fig). Western blot analysis of larval lysates using anti-Asap antibody detected a predicted band of ~130 kDa in control animals, while this band was not detectable in the *Asap^K23/B52* larvae (Fig 1B and 1H).

We tried to determine the localization of the Asap protein. Two antibodies raised against Asap [amino acids 1–412 (this work) or 754–1092 [22] did not specifically detect the endogenous Asap: pre-absorption of the antibody with pure Asap protein eliminated all staining. This suggested the presence of similar epitopes/proteins at the NMJ (Fig 1B and S2 Fig). However, neuronal overexpression of an *Asap-mCherry* transgene [23] gets targeted to the NMJ (S2 Fig), supporting its possible requirement at the synaptic sites.

Compared to the control NMJs, *Asap^K23/B52* mutant NMJs showed a significant reduction in average bouton number, increased average bouton area, increased average bouton size, and increased maximum inter-bouton diameter (Fig 1C-1N). The aberrant synaptic morphology of *Asap* mutants was restored to the control level by expressing a full-length *Asap* transgene in the motor neurons or ubiquitously in *Asap^K23/B52* (Fig 1E-1F and 1K-1N). However, muscle-specific expression of the *Asap* transgene in the *Asap^K23/B52* did not restore the NMJ defects (Fig 1G and 1K-1N). Furthermore, neither UAS-Asap nor D42-Gal4 alone in *Asap* mutant background rescued the phenotypes, confirming that the observed defects arise from loss of Asap function rather than insertional or leaky transgene effects (S3 Fig). Together, these data suggest that Asap is required in neurons to regulate NMJ bouton morphology.

Since mammalian ASAP1 modulates cytoskeletal dynamics [24,25], we next asked if the neuronal cytoskeleton was altered due to the loss of *Asap*. Using an antibody against microtubule-associated protein 1B (MAP1B/Futsch), we found a significant reduction in the percentage of Futsch-positive loops in *Asap^K23/B52* synapses (Fig 1O-1P). The reduced

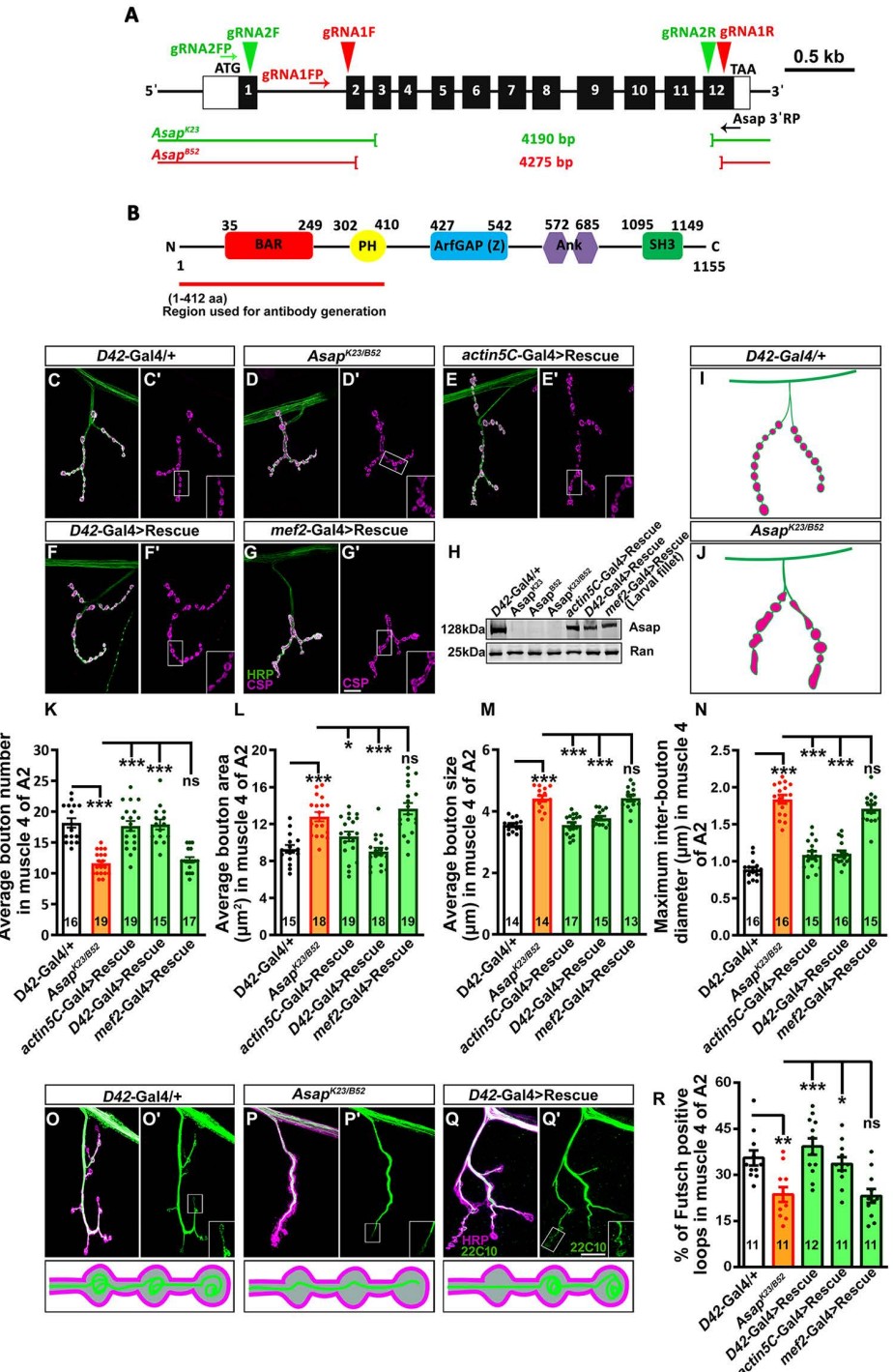

Fig 1. Alterations in synaptic morphology and organization in *Asap* mutants. (A) Schematic representation of the *Asap* locus showing exons (solid black boxes, 1-12) and introns (thin black lines). Red and green arrowheads represent gRNA target sites to generate *Asap* alleles. The first gRNA (red arrowhead) and second gRNA (green arrowhead) were designed for the 2nd and 12th and 1st and 12th exons of the *Asap* locus, respectively. Mutant lines were screened using Asap 5′ FP and Asap 3′ RP primers (black arrows). The thin green and red lines represent *Asap*$^{K23}$ and *Asap*$^{B52}$ deletion alleles. (B) Domain organization of the Asap protein. The N-terminal 1-412 amino acid was used to generate antibodies against Asap. (C-G′) Confocal images of NMJ synapses at muscle 4 of A2 hemisegment showing synaptic growth in (C-C′) *D42-Gal4/* +control, (D-D′) *Asap*$^{K23/B52}$ heteroallelic mutant, (E-E′) *actin5C-Gal4* driven rescue line (*actin5C-Gal4, Asap*$^{K23/B52}$; *UAS-Asap*$^{FL}$/+), (F-F′) *D42*-Gal4 driven rescue line (*Asap*$^{K23/B52}$; *D42-Gal4/UAS-Asap*$^{FL}$)

and (G-G′) *mef2*-Gal4 driven rescue line (*Asap^K23/B52*; *mef2-Gal4/UAS-Asap^FL*) double immunolabeled for neuronal membrane marker, HRP (green) and presynaptic vesicle marker, CSP (magenta). The scale bar in G′ for (C-G′) represents 20 μm. **(H)** Western blot showing protein levels of Asap in controls, homozygous *Asap^K23*, homozygous *Asap^B52*, heteroallelic *Asap^K23/B52*, *actin5C*-Gal4 driven rescue, *D42*-Gal4 driven rescue, and *mef2* Gal4 driven rescue animals. Ran protein levels were used as an internal loading control. **(I, J)** Schematic representation of NMJ in control (I) and *Asap* mutant (J) animals as indicated. **(K-N)** Histogram showing an average number of boutons **(K)**, average bouton area **(L)**, average bouton size **(M)**, and maximum interbouton diameter (N) at muscle 4 NMJ of A2 hemisegment of the indicated genotypes. *$p$ = 0.05, ***$p$ = 0.001; ns, not significant. The statistical analysis was done using one-way ANOVA followed by post-hoc Tukey's multiple-comparison test. n = 14-19 NMJ per genotype. All values represent mean ± SEM. **(O-Q)** Confocal images of NMJ synapses at muscle 4 of A2 hemisegment showing futsch loops in indicated genotypes double immunolabeled with 22C10 (green) and HRP (magenta). The lower panels show a schematic representation of Futsch loops in the indicated genotypes. Scale bar in Q′ (for O-Q′) represents 4 μm. **(R)** Histogram showing the percentage of the Futsch positive loops from muscle 4 NMJ at A2 hemisegment in the indicated genotypes. *$p$ = 0.05, **$p$ = 0.01, ***$p$ = 0.001; ns, not significant. The statistical analysis was done using one-way ANOVA followed by post-hoc Tukey's multiple-comparison test. n = 11-12 NMJ per genotype. All values represent mean ± SEM. The cartoons in panels (I-J, O-Q lower panels) are created using Adobe Illustrator. The values for each quantification are shown in Table A in S1 Text.

microtubule loops in mutants were rescued to the control level by expressing a full-length Asap transgene in the motor neurons of *Asap^K23/B52* (Fig 1O-1R). In contrast, western blot analysis showed that total α-tubulin or acetylated-tubulin levels were unaltered in the *Asap* mutants, indicating that overall microtubule abundance was not affected. Furthermore, loss of *Asap* did not alter levels of synaptic actin or the number of moesin punctae, suggesting that the microfilament-based cytoskeleton remains intact in *Asap^K23/B52* mutants (S4 Fig). Together, these data indicate a crucial role of Asap in regulating bouton morphology, likely through the regulation of neuronal microtubule organization.

### Asap regulates neurotransmitter release and SV release probability

We measured the spontaneous and evoked synaptic potentials to determine whether the altered NMJs due to the loss of *Asap* were accompanied by functional changes. We found that while the amplitude of spontaneous miniature excitatory postsynaptic potentials (mEPSP) was not significantly different from wild-type controls, the miniature frequency and excitatory postsynaptic potentials (EPSP) were significantly increased in *Asap^K23/B52* mutant NMJs (Fig 2A-2F). However, *Asap^K23/B52* NMJs showed no significant change in the quantal content (QC), an estimation of the number of vesicles released per action potential, compared to control NMJs (Fig 2G). The miniature frequency and evoked amplitude phenotypes in *Asap^K23/B52* were significantly rescued by expressing *Asap* transgene in the motor neurons (Fig 2E-2F).

The increased mEPSP frequency and EPSP amplitude could arise due to altered active zone numbers or changes in the organization of glutamate receptors at synapses [26]. To test these possibilities, we labeled the *Asap^K23/B52* NMJs with anti-Bruchpilot (Brp) and anti-GluRIII antibodies to quantify the number of the presynaptic active zone and postsynaptic glutamate receptor clusters, respectively. We found a significant increase in Brp punctae per NMJ in *Asap^K23/B52* compared to the control NMJs (average number of Brp punctae/NMJ: $p$ = 0.0001). Moreover, we found increased Brp puncta density at the mutant NMJ compared to controls (Fig 2H-2I "and 2K-2L). Additionally, the area of the GluR cluster was increased in the mutants compared to control animals, and the receptors were tightly apposed to the corresponding Brp-positive active zone (Fig 2H-2I"). The enhanced Brp density and GluRIII receptor clusters in mutants were restored to the control level by expressing a *Asap* transgene in the motor neurons of *Asap* mutants (Fig 2H-2M).

To test whether alterations in active zone structure or voltage-gated calcium channel (VGCC) activity contribute to the *Asap* mutant phenotype, we quantified Brp and Cacophony (CAC) intensities in control, *Asap* mutant, and rescue conditions. We observed approximately a 45% reduction in Brp intensity and a 50% reduction in CAC intensity in *Asap* mutants, accompanied by a concurrent increase in AZ density, measured as the number of Brp or CAC puncta per NMJ, relative to controls (Fig 2N-2V). These data indicate that (a) *Asap* may influence the nanoscale organization of active zones, and (b) reductions in Brp intensity may be compensated for by increased AZ density as part of a compensatory, homeostatic mechanism [27–29].

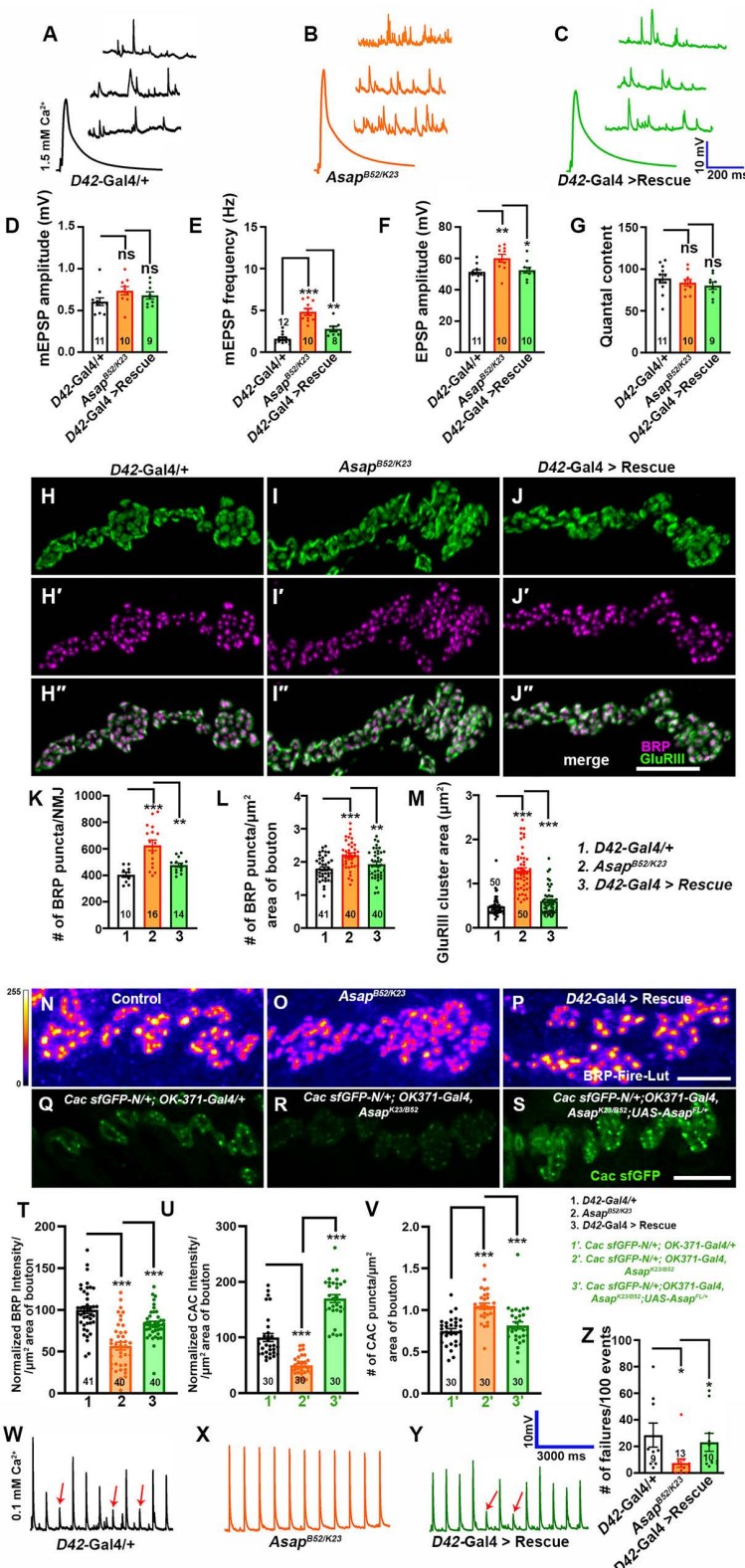

**Fig 2. Gain in synaptic functions and reduced synaptic failures in *Asap* mutants. (A-C)** Representative traces of mEPSP and EPSP in **(A)** *D42-Gal4/* +control, **(B)** *Asap^{K23/B52}*, and **(C)** *Asap^{K23/B52}; D42-Gal4/UAS-Asap^{FL}* larvae. **(D-G)** Histogram showing mEPSP amplitude **(D)**, mEPSP frequency

**(E)**, EPSP amplitude **(F)**, and quantal content (G) from muscle 6 of A2 hemisegment in *D42-Gal4/ +*control, *Asap*[K23/B52] mutant and *Asap*[K23/B52]; *D42-Gal4/ UAS-Asap*[FL] animals. ns, not significant. *$p = 0.05$, **$p = 0.001$, ***$p = 0.0001$; ns, not significant. The statistical analysis was done using one-way ANOVA followed by post-hoc Tukey's multiple-comparison test. n = 8-13 NMJ per genotype. All values represent mean ± SEM. **(H-J")** Confocal images of NMJ synapses at muscle 4 of A2 hemisegment of (H-H") *D42-Gal4/+* control, (I-I") *Asap*[K23/B52] mutants and (J-J") *Asap*[B52/K23]; *D42-Gal4/UAS-Asap*[FL] double immunolabeled with antibodies against active zones marker Bruchpilot, Brp (magenta) and GluRIII (green). The scale bar in J" represents 2.5 μm. **(K)** Histogram showing the number of Brp punctae per NMJ from muscle 4 at A2 hemisegment in *D42-Gal4/ +*control, *Asap*[K23/B52] and *Asap*[K23/B52]; *D42-Gal4/ UAS-Asap*[FL] animals. **$p = 0.001$, ***$p = 0.0001$. n = 10-16 NMJ per genotype. **(L)** Histogram showing the number of Brp punctae per μm² area of bouton from muscle 4 NMJ at A2 hemisegment in control, *Asap*[K23/B52] and *Asap*[K23/B52]; *D42-Gal4/UAS-Asap*[FL] animals. **$p = 0.001$, ***$p = 0.0001$. n = 40-41 boutons per genotype. **(M)** Histogram showing the area of the GluRIII cluster from muscle 4 NMJ at A2 hemisegment in *D42-Gal4/ +*control, *Asap*[K23/B52], and *Asap*[K23/B52]; *D42-Gal4/UAS-Asap*[FL] animals. ***$p = 0.0001$. n = 50 boutons per genotype. The statistical analysis for K-M was done using one-way ANOVA followed by post-hoc Tukey's multiple-comparison test. All values represent mean ± SEM. **(N-P)** Confocal images of NMJ synapses at muscle 4 of A2 hemisegment of **(N)** *D42-Gal4/ +*control, **(O)** *Asap*[K23/B52] mutants and **(P)** *Asap*[B52/K23]; *D42-Gal4/UAS-Asap*[FL] immunolabeled with antibodies against active zones marker Bruchpilot, Brp (Fire-Lut). The scale bar in P represents 2.5 μm. **(Q-S)** Confocal images of NMJ synapses at muscle 4 of A2 hemisegment of **(Q)** *Cac sfGFP-N/ +; OK-371-Gal4/+* **(R)** *Cac sfGFP-N/ +; OK371-Gal4, Asap*[K23/B52] and **(S)** *Cac sfGFP-N/ +; OK371-Gal4, Asap*[K23/B52]; *UAS-Asap*[FL]/ +immunolabeled with nanobodies against GFP, which labels endogenous CAC sfGFP (green) to mark voltage-gated calcium ion channels. The scale bar in P represents 2.5 μm. **(T-V)** Histogram showing Brp intensity **(T)**, CAC intensity **(U)** and CAC density **(V)** per μm² area of bouton from muscle 4 NMJ at A2 hemisegment in indicated genotypes. ***$p = 0.0001$. n = 10-60 boutons per genotype. **(W-Y)** Representative traces of action potential firing at 0.1 mM extracellular Ca²⁺ in **(W)** *D42-Gal4/ +*control, **(X)** *Asap*[K23/B52], **(Y)** *Asap*[K23/B52]; *D42-Gal4/UAS-Asap*[FL] larvae. Scale bar for EPSPs is x = 3000 ms and y = 10 mV. **(Z)** Histogram showing the percentage of failures in *D42-Gal4/ +*control, *Asap*[K23/B52], *Asap*[K23/B52]; *D42-Gal4/UAS-Asap*[FL] animals. *$p = 0.05$. n = 9-13 NMJ per genotype. The statistical analysis was done using one-way ANOVA followed by post-hoc Tukey's multiple-comparison test. All values represent mean ± SEM. The values for each quantification are shown in Table B in S1 Text.

Given the increased number of Brp puncta in *Asap* mutants, we hypothesized that the mutant synapse could sustain synaptic vesicle release under low extracellular Ca²⁺. To test this hypothesis, we used failure analysis to determine the number of failed release events at low (0.1 mM) extracellular Ca²⁺ concentrations. We observed a significantly lower failure probability in *Asap*[K23/B52] compared to the control NMJs (Fig 2W-2Z). Consistent with these observations, loss of *Asap* showed reduced paired-pulse facilitation (PPF), indicating an increased release probability at *Asap* mutant NMJs under low calcium concentrations [30] (S5 Fig). Together, these data suggest that loss of *Asap* results in increased synaptic vesicle release probability and gains of neurotransmission, possibly due to increased active zone numbers.

Elevated synaptic release can occur in a variety of ways, including engaging neuronal machinery that triggers forms of presynaptic homeostatic potentiation (PHP), elevated calcium ions, or both [31,32]. Hence, we tested whether loss of *Asap* induces PHP or alters intracellular calcium levels.

## Presynaptic homeostatic plasticity remains intact in *Asap* mutant

If *Asap* loss were normally engaging the PHP machinery, then a dual combination of *Asap* loss plus a PHP challenge would likely occlude any further increases in release. To assess the rapid induction of PHP, we utilized *Asap*[K23/B52] in conjunction with Philanthotoxin-433 (PhTx). PhTx selectively inhibits postsynaptic glutamate receptors, reducing mEPSP amplitudes while increasing QC to sustain evoked synaptic potentials at near-normal levels [33,34] (Fig 3A-3B). PhTx treatment decreased mEPSP amplitude and frequency relative to untreated NMJs in the controls and *Asap* mutants (Fig 3C-3J). Moreover, PhTx-treated control and *Asap* mutant showed a significant increase in the QC relative to untreated animals (Fig 3C-3J). Quantitative analysis revealed an approximately 63% increase in QC in PhTx-treated controls and a ~ 33% increase in PhTx-treated Asap mutants compared with their respective untreated conditions (Fig 3C-3J). As a result of these increases in QC, PhTx-treated controls and PhTx-treated *Asap* mutants both showed evoked NMJ excitation (EPSP) levels that were indistinguishable from non-treated controls (Fig 3I). This result indicates fully intact PHP in *Asap* mutants, and therefore, these data suggest that loss of *Asap* likely enhances NMJ function in a manner largely independent of PHP signaling paradigms.

Since rapid induction of PHP is also associated with an accumulation of the active zone protein Brp at the nerve terminal [35], we further tested whether acute PhTx application would similarly induce Brp signal accumulation in *Asap* mutant.

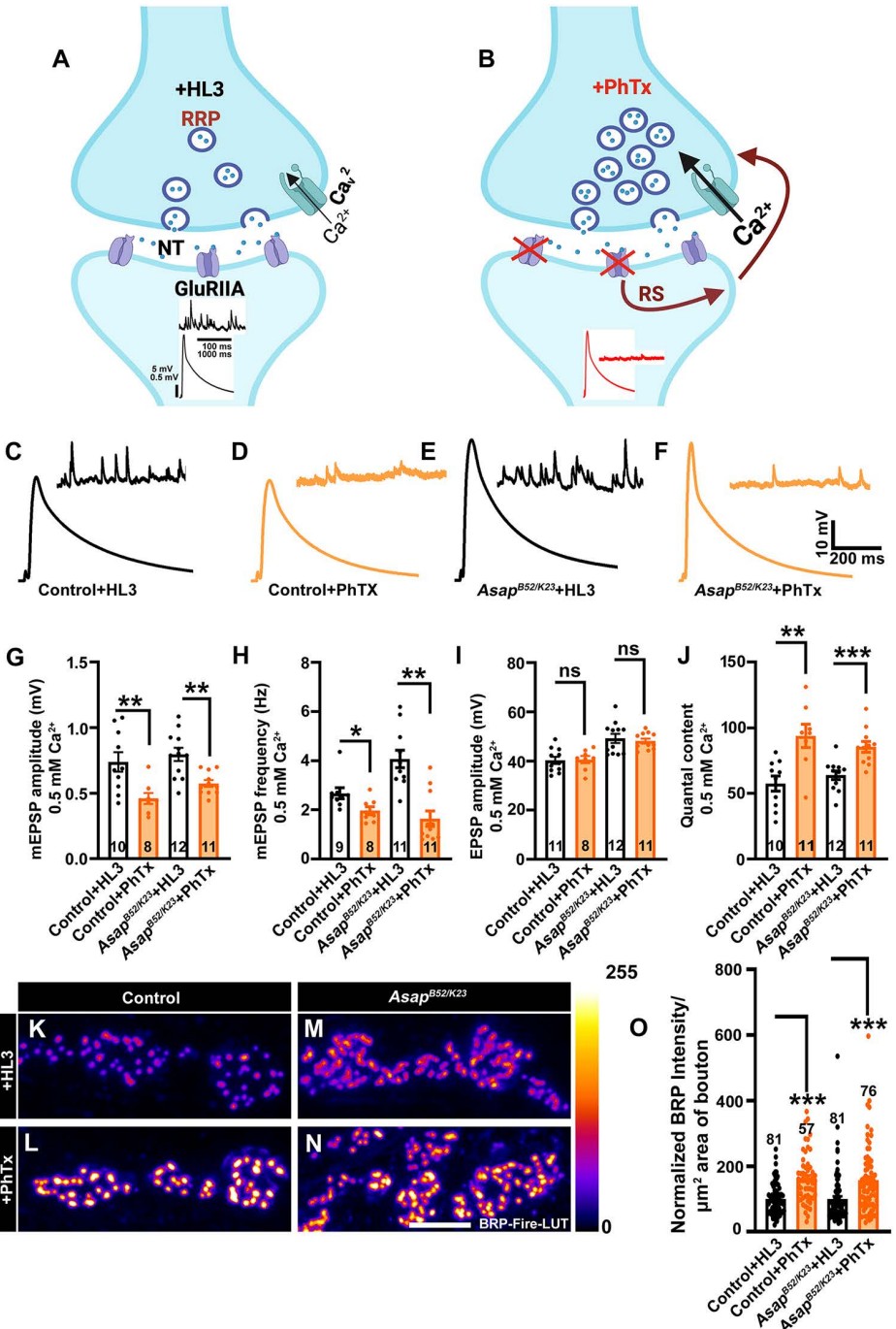

**Fig 3. Homeostatic synaptic plasticity is intact at *Asap* mutant NMJs. (A-B)** A schematic model illustrating the homeostatic regulation of neurotransmitter release at the nerve terminal. Pharmacological or genetic inhibition of postsynaptic glutamate receptors leads to a marked reduction in quantal size, as reflected by diminished amplitudes of spontaneous miniature excitatory postsynaptic potentials (small traces). Despite this decrease, evoked excitatory postsynaptic potentials (large traces) remain relatively unaffected due to a homeostatic retrograde signaling mechanism (RS, dark orange arrows) originating from the muscle and targeting the presynaptic terminal. This compensatory response enhances neurotransmitter release by increasing quantal content through at least two mechanisms: (1) an elevated presynaptic calcium influx, likely mediated by $Ca_V2$-type voltage-gated calcium channels, and (2) increase in the readily releasable pool of synaptic vesicles [82]. Cartoon in panel A and B created in BioRender; (Mallik, B. (2025) https://BioRender.com/dpwo7xr). **(C-F)** Representative traces of mEPSPs and EPSPs in (C) control+HL3, (D) control+PhTx, **(E)** *Asap^B23/K23*+HL3 and **(F)** *Asap^K23/B52*+PhTx larvae. **(G-J)** Histogram showing (G) mEPSP amplitude, (H) mEPSP frequency, **(I)** EPSP amplitude and **(J)** Quantal content

normalized to untreated animals from muscle 6 of A2 hemisegment in control + HL3, control + PhTx, $Asap^{K23/B52}$ + HL3 mutant and $Asap^{K23/B52}$ + PhTx animals. ns, not significant. **$p = 0.007$, **$p = 0.001$ (mEPSP amplitude), *$p = 0.02$, **$p = 0.0001$ (mEPSP frequency), **$p = 0.002$, ***$p = 0.0005$ (quantal content); ns, not significant. The statistical analysis was done using Student's t-test for the pairwise comparison. n = 8-12 NMJ per genotype. All values represent mean ± SEM. **(K-N)** Confocal images of NMJ synapses at muscle 6/7 of A2 hemisegment of (K) control + HL3, (L) control + PhTx, **(M)** $Asap^{K23/B52}$ + HL3 mutants and **(N)** $Asap^{K23/B52}$ + PhTx immunolabeled with antibodies against active zones marker Bruchpilot, Brp (Brp fluorescence is represented as LUT). The scale bar in N (K-N) represents 2.5 μm. **(O)** Histogram showing the normalized Brp intensity per μm² area of bouton from muscle 6/7 at A2 hemisegment in control + HL3, control + PhTx, $Asap^{K23/B52}$ + HL3 and $Asap^{K23/B52}$ + PhTx treated animals. ***$p = 0.0001$. The statistical analysis was done using Student's t-test for the pairwise comparison. n = 57-81 boutons per genotype. All values represent mean ± SEM. The values for each quantification are shown in Table C in S1 Text.

Indeed, we observed elevated Brp signal in PhTx-treated control as well as *Asap* mutant NMJs compared to untreated fillets (Fig 3K-3O). These findings suggest that PHP is intact in the *Asap* mutant and that the gain in synaptic transmission due to the loss of *Asap* is not due to pre-induced PHP. Next, we considered if calcium levels were affected in Asap-deficient synapses.

## Asap maintains synaptic activity by regulating calcium release from the ER

Increased resting synaptic calcium levels could facilitate the gain in neurotransmitter release [36,37] observed in the *Asap* mutants. Using the genetically encoded calcium sensor tdTomato-P2A-GCaMP5G [36], we checked if loss of *Asap* leads to elevated resting synaptic calcium [$Ca^{2+}$]. Indeed, we found an elevated GCaMP5G to tdTomato fluorescence ratio, indicating higher resting synaptic calcium [$Ca^{2+}$] levels in the *Asap*-deficient synapses (Fig 4A-4E). To further test if the elevated synaptic calcium in the *Asap* mutant was associated with gains in synaptic transmission, we chelated the intracellular calcium using membrane-permeant BAPTA-AM. Indeed, we found that chelating calcium intracellularly suppressed evoked activity and miniature frequencies in *Asap* mutants compared to the DMSO-treated mutant animals (Fig 4F-4L). To assess whether elevated presynaptic calcium levels are associated with the increased EPSP amplitudes observed in *Asap* mutants, we measured activity-regulated live calcium responses using the calcium indicator GCaMP5G. NMJs were stimulated with HL3 buffer containing 90 mM KCl and 0.5 mM $CaCl_2$. Under these conditions, control boutons exhibited an approximately 30% increase in GCaMP5G fluorescence relative to unstimulated boutons. In contrast, *Asap* mutant boutons exhibited enhanced response, with an approximately 100% increase relative to their respective unstimulated controls (S6 Fig). Although, high KCl stimulation induces global depolarization, the enhanced calcium response in the *Asap* mutant bouton is consistent with increased presynaptic calcium during synaptic activation. Together, these results support the idea that elevated calcium release from intracellular stores, enhanced activity-induced calcium response, and increased active zone numbers in *Asap* mutants contribute to maintaining robust neurotransmission.

Next, we genetically perturbed IP3R, RyR, or Letm1 to identify the pathway for the elevated resting synaptic [$Ca^{2+}$]. Consistent with prior reports, we found that genetic perturbation of IP3R activity (blocked using neuronally expressed IP3-Sponge.m30 [8,38] or RyR activity (depleted using neuronally expressed *RyR* RNAi) [8,39] did not show significant changes in the basal or evoked synaptic activity (Fig 4U-4V). However, neuronally blocking IP3R activity or RyR in the *Asap* mutant suppressed the enhanced mEPSP frequency phenotype (Fig 4O-4R and 4U-4V). Moreover, neuronally blocking IP3R and RyR in *Asap* mutants suppressed the evoked activity to the control levels (Fig 4S and 4U-4V). To directly assess whether elevated ER-dependent synaptic calcium release underlies the gain in synaptic functions observed in *Asap* mutants, we neuronally knocked down *itpr* or *RyR* in *Asap* mutant background. Knockdown of either *itpr* or *RyR* suppressed the elevated synaptic calcium observed in *Asap* mutants (S7 Fig). Simultaneously knocking down *itpr* and blocking *RyR* also suppressed the elevated calcium in *Asap* mutants (S7 Fig). Together, these data indicate that Asap regulates resting synaptic [$Ca^{2+}$] through IP3 and RyR pathways.

To strengthen our observation that the elevated spontaneous release observed in *Asap* mutants arises from enhanced calcium release from internal ER stores, we examined *Asap* mutants in heterozygous *itpr* and *RyR* mutant backgrounds.

**Fig 4. Asap modulates synaptic activity by controlling cytoplasmic calcium levels at the nerve terminals. (A-B'')** Confocal live images of NMJ synapses at muscle 6/7 of the A2 hemisegment of (A-A'') control and (B-B'') $Asap^{K23/B52}$ mutant expressing GCaMP5G (green) and tdTomato (red) fluorescent protein. The scale bar in B'' for (A-B'') represents 4 µm. **(C-D)** Intensity plot profile for GCaMP5G (green) and tdTomato (red) across the bouton (shown in A'' and B'' and thin line). **(E)** Histogram showing the fluorometric ratio of GCaMP5G and tdTomato per µm² bouton area in control (100.0±8.57) and $Asap^{K23/B52}$ (194.3±25.96) larvae. **p = 0.004. The statistical analysis was done using Student's t-test for the pairwise comparison. n = 16-21

boutons per genotype. All values represent mean ± SEM. **(F)** A schematic model depicting the role of calcium release from endoplasmic reticulum via IP3 receptors (IP3Rs) and ryanodine receptors (RyRs) in synaptic transmission. The model includes key components such as mitochondria, Ca$_v$2-type voltage-gated calcium channels, and synaptic vesicles in $Asap^{K23/B52}$ synaptic terminals treated with BAPTA-AM. **(G-J)** Representative traces of mEPSP and EPSP in (G) control + DMSO, **(H)** $Asap^{K23/B52}$ + DMSO, **(I)** Control + BAPTA-AM, and **(J)** $Asap^{K23/B52}$ + BAPTA-AM treated larvae. Scale bars for EPSPs (mEPSP) are x = 50 ms (1000 ms) and y = 10 mV (1 mV). **(K-L)** Histogram showing mEPSP frequency **(K)**, and EPSP amplitude (L) from muscle 6 of A2 hemisegment in control + DMSO, $Asap^{K23/B52}$ + DMSO, control + BAPTA-AM, and $Asap^{K23/B52}$ + BAPTA-AM treated animals. *$p$ = 0.01, **$p$ = 0.002, ***$p$ = 0.0003 (mEPSP amplitude), *$p$ = 0. 01, **$p$ = 0.006, *** $p$ = 0.0003 (EPSP amplitude). The statistical analysis was done using Student's t-test for the pairwise comparison. n = 6-9 NMJ per genotype. All values represent mean ± SEM. **(M-N)** A schematic model illustrating IP3 receptors (IP3Rs) and ryanodine receptors (RyRs) induced calcium release from the ER in synaptic transmission. The model highlights key components, including mitochondria, Ca$_v$2-type voltage-gated calcium channels, and synaptic vesicles, in $Asap^{K23/B52}$ mutant synaptic terminals (M) and those subjected to combined RyR and IP3R blockade **(N)**. **(O-T)** Representative traces of mEPSP and EPSP in **(O)** $D42$-$Gal4/$ + control, **(P)** $Asap^{K23/B52}$, **(Q)** $Asap^{K23/B52}$ + $D42$-$Gal4 > UAS$-$ip3$-$sponge.m30$, **(R)** $Asap^{K23/B52}$ + $D42$-$Gal4 > UAS$-$RyR$ $RNAi$, **(S)** $Asap^{K23/B52}$ + $D42$-$Gal4 > UAS$-$RyR$ $RNAi$ + $D42$-$Gal4 > UAS$-$ip3$-$sponge.m30$, and **(T)** $Asap^{K23/B52}$ + $D42$-$Gal4 > UAS$-$Letm1$ $RNAi$ larvae. Scale bars for EPSPs (mEPSP) are x = 50 ms (1000 ms) and y = 10 mV (1 mV). **(U-V)** Histogram showing mEPSP frequency (U) and EPSP amplitude (V) from larval muscle 6 of A2 hemisegment in $D42$-$Gal4/$ + control, $Asap^{K23/B52}$, $Asap^{K23/B52}$ + $D42$-$Gal4 > UAS$-$ip3$-$sponge.m30$, $Asap^{K23/B52}$ + $D42$-$Gal4 > UAS$-$RyR$ $RNAi$, $Asap^{K23/B52}$ + $D42$-$Gal4 > UAS$-$RyR$ $RNAi$ + $D42$-$Gal4 > UAS$-$ip3$-$sponge.m30$, and $Asap^{K23/B52}$ + $D42$-$Gal4 > UAS$-$Letm1$ $RNAi$ larvae. ***$p$ = 0.0001, **$p$ = 0.0004, **$p$ = 0.0002, **$p$ = 001; ns, not significant (for mEPSP frequency), ***$p$ = 0.0001, **$p$ = 0.0001; ns, not significant (for EPSP amplitude). The statistical analysis was done using one-way ANOVA followed by post-hoc Tukey's multiple-comparison test. n = 8-14 NMJ per genotype. All values represent mean ± SEM. The cartoons in panels (F, M-N) are created using Adobe Illustrator. The values for each quantification are shown in Table D in S1 Text.

Both $itpr^{90B.0}/$ +; $Asap^{B52/K23}$ and $RyR^{E4340K}/$ +; $Asap^{B52/K23}$ combinations displayed a mild but consistent reduction in mEPSP frequency and a significant decrease in EPSP amplitude compared with $Asap$ mutants alone (S8 Fig). These partial suppressions suggest that calcium release through IP3 and ryanodine receptors contributes to the elevated neurotransmitter release in $Asap$ mutants.

Next, to assess possible Asap-regulated calcium flux from mitochondria, we depleted the $Letm1$ in $Asap$ mutant motor neurons; however, we did not observe any notable change in mini frequencies and evoked responses compared to $Asap$ mutants (Fig 4T-4V). These data suggest calcium release from ER through the IP3-dependent pathway underlies elevated resting synaptic calcium levels in the $Asap$-deficient synapses.

### Asap suppresses the PLCβ-dependent calcium release pathway

The $Plc21C$ gene, encoding the $Drosophila$ homolog of PLCβ, is a component of the IP3 signaling pathway, and it is essential for maintaining presynaptic homeostatic potentiation [8,40]. While our data indicate that IP3 sequestration suppresses heightened neurotransmission in $Asap$ mutants, the specific contribution of PLCβ signaling to synaptic activity remains unclear (Fig 5A), prompting further investigation into its role in regulating miniature frequency and evoked neurotransmission. To test whether the elevated presynaptic calcium levels in $Asap$ mutants depend on PLCβ signaling, we expressed $Plc21C$ RNAi in the $Asap$ mutant background. Although PLCβ knockdown alone did not significantly alter GCaMP5G fluorescence intensity, its knockdown in $Asap$ mutants significantly suppressed the elevated GCaMP5G signal (Fig 5B-5J), indicating that PLCβ activity contributes to the elevated synaptic calcium in $Asap$ mutants. Consistently, neuronal depletion of $Plc21C$ in $Asap$ mutant significantly reduced both mEPSP frequency and EPSP amplitude to control levels (Fig 5K-5R). Together, these findings suggest that Asap acts upstream of the PLCβ-IP3 signaling pathway to modulate intracellular calcium dynamics and normal synaptic functions.

### Asap attenuates Arf6-dependent signaling to regulate presynaptic morphogenesis

$Drosophila$ Asap regulates Arf1 and Arf6 through its conserved ArfGAP domain [20,22,23]. Since the ArfGAP domain negatively regulates Arf.GTP levels, we hypothesized that loss of $Asap$ may lead to heightened Arf activity.

To test this directly, we expressed dominant negative Arf6 (Arf6$^{DN}$) in $Asap$ mutant background. Neuronal ($Asap^{K23/B52}$; $D42$-$Gal4 > UAS$-$Arf6^{DN}$) or ubiquitous ($Asap^{K23/B52}$; $actin5C$-$Gal4 > UAS$-$Arf6^{DN}$) expression restored the NMJ structural defects of $Asap$ mutants, yielding normal numbers of Futsch loops as well as normal Brp number and density (Figs 6C-6P

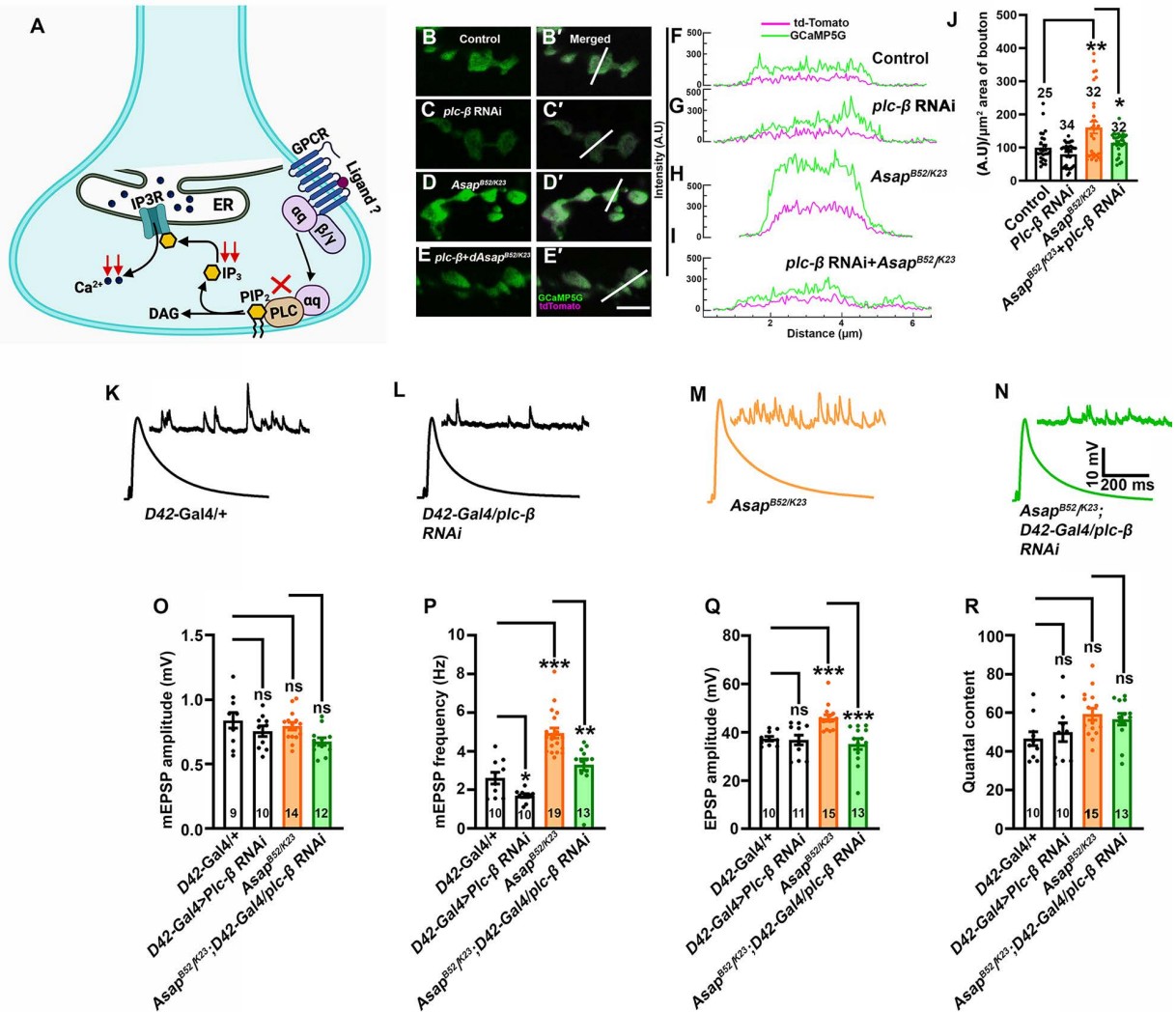

**Fig 5.** ***Asap* mutation induces IP3-mediated ER-calcium release via modulation of PLCβ signaling. (A)** A schematic model illustrating G protein-coupled receptor (GPCR)-mediated IP3-dependent signaling in neurons, leading to calcium release from the ER. Activation of GPCRs triggers phospholipase C (PLC), which catalyzes the production of inositol 1,4,5-trisphosphate (IP3). IP3 binds to IP3 receptors (IP3Rs) on the ER membrane, facilitating calcium release into the cytoplasm. Key components involved in this signaling cascade, including GPCRs, PLC, IP3, IP3Rs, and intracellular calcium dynamics, are depicted. Cartoon in panel A created in BioRender; (Mallik, B. (2025) https://BioRender.com/e590i4r). **(B-E')** Confocal live images of NMJ synapses at muscle 6/7 of A2 hemisegment of (B-B') *OK371-Gal4/ +; UAS- td-Tomato-GCaMP5G/+* (control), (C-C') *OK371-Gal4/ +; UAS-plc-β RNAi/UAS-td-Tomato-GCaMP5G,* (D-D') *OK371-Gal4, Asap^{B52/K23}; UAS- td-Tomato-GCaMP5G/+* and (E-E') *OK371-Gal4, Asap^{B52}/ OK371-Gal4, Asap^{K23}; UAS-td-Tomato-GCaMP5G/UAS-plc-β RNAi* expressing GCaMP5G (green) and td-Tomato (magenta) fluorescent protein. The scale bar in E' for (B-E') represents 4 μm. **(F-I)** Intensity plot profile for GCaMP5G (green) and td-Tomato (magenta) across the bouton (shown in B', C', D' and E' as a thin line). **(J)** Histogram showing the fluorometric ratio of GCaMP5G and tdTomato per μm² bouton area in the indicated genotypes. **\*\****p = 0.007, \*p = 0.016. The statistical analysis was done using one-way ANOVA followed by post-hoc Tukey's multiple-comparison test. n = 32-34 boutons per genotype. All values represent mean ± SEM. **(K-N)** Representative traces of mEPSP and EPSP in **(K)** *D42-Gal4/ +* control, **(L)** *D42-Gal4 > UAS-plc-β RNAi*, **(M)** *Asap^{K23/B52}*, and **(N)** *Asap^{K23/B52}; D42-Gal4 > UAS-plc-β RNAi* animals. Scale bars for EPSPs (mEPSP) are x = 200 ms (1000 ms) and y = 10 mV (1 mV). **(O-R)** Histogram showing mEPSP amplitude **(O)**, mEPSP frequency **(P)**, EPSP amplitude **(Q)**, and quantal content (R) from muscle 6 of A2 hemisegment in *D42-Gal4/ +* control, *D42-Gal4 > UAS-plc-β RNAi*, *Asap^{K23/B52}*, and *Asap^{K23/B52}+ D42-Gal4 > UAS-plc-β RNAi* larvae. ns; not significant, \*p = 0.01, \*\*p = 0.0003, \*\*\*p = 0.0001 (for mEPSP frequency), \*\*\*p = 0.0001, \*\*\*p = 0.0002). The statistical analysis was done using one-way ANOVA followed by post-hoc Tukey's multiple-comparison test. n = 9-19 NMJ per genotype. All values represent mean ± SEM. The values for each quantification are shown in Table E in S1 Text.

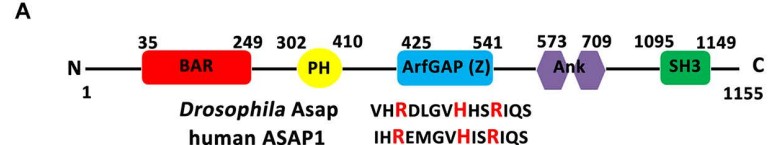

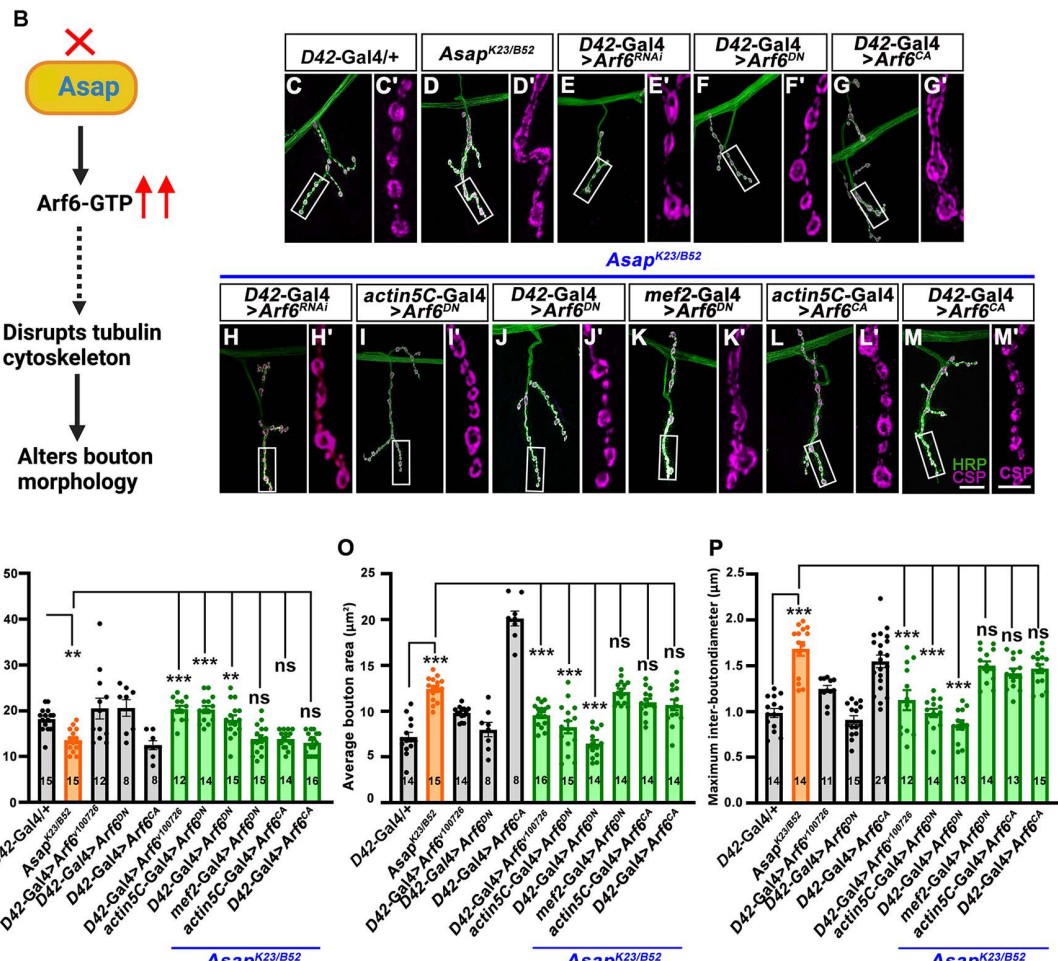

HRP
CSP

CSP

**Fig 6. Asap modulates Arf6-dependent signaling to regulate presynaptic morphogenesis. (A)** A schematic representation of Asap shows that it contains various conserved domains that can regulate membrane dynamics and Arf-dependent activities. The amino acid residues of the ArfGAP domain (labeled in red) are conserved in humans and *Drosophila*. **(B)** The pathway illustrates how Asap might modulate ARF6.GTP levels to stabilize the tubulin cytoskeleton and regulate bouton morphology. In *Asap* null mutants, the absence of the ArfGAP domain prevents ARF6.GTP hydrolysis, leading to its accumulation in the cytoplasm. This dysregulation disrupts the tubulin cytoskeleton, resulting in altered bouton morphology. **(C-M′)** Confocal images of NMJ synapses at muscle 4 of A2 hemisegment showing synaptic growth in (C, C′) *D42-Gal4/+*control, (D, D′) *Asap^{K23/B52}*, (E, E′) *UAS-Arf6^{RNAi}/+; D42-Gal4/+*, (F, F′) *UAS-Arf6^{DN}/D42-Gal4*, (G, G′) *UAS-Arf6^{CA}/D42-Gal4*, (H, H′) *UAS-Arf6^{RNAi}, Asap^{K23/B52}; D42-Gal4/+*, (I, I′) *actin5C-Gal4, Asap^{K23/B52}; UAS-Arf6^{DN}/+*, (J, J′) *Asap^{K23/B52}; D42-Gal4/UAS-Arf6^{DN}*, (K, K′) *Asap^{K23/B52}; mef2-Gal4/UAS-Arf6^{DN}*, (L, L′) *actin5C-Gal4, Asap^{K23/B52}; UAS-Arf6^{CA}/+*, and (M, M′) *Asap^{K23/B52}; D42-Gal4/UAS-Arf6^{CA}* double immunolabeled for CSP (magenta) and HRP (green). Scale bar in M (for merged images, C-M) and M′ (for C′-M′) represents 10 μm. **(N-P)** Histogram showing an average number of boutons **(N)**, average bouton area **(O)**, and maximum interbouton diameter (P) at the muscle 4 NMJ of the A2 hemisegment of the indicated genotypes. ns; not significant, \*\**p* = 0.0009 (bouton number), \*\*\**p* = 0.0001. The statistical analysis was done using one-way ANOVA followed by post-hoc Tukey's multiple-comparison test. n = 8-16 NMJ per genotype. All values represent mean ± SEM. The values for each quantification are shown in Table F in S1 Text.

and S9). Similar results were obtained when Arf6 was depleted in *Asap*<sup>K23/B52</sup> motor neurons by expressing an *Arf6* RNAi (Fig 6C-6P). By contrast, expression of *Arf6*<sup>DN</sup> in muscle did not restore the synapse morphology (Fig 6K and 6N–6P). Hence, we did not investigate the Brp number and density in the muscle-specific expression of *Arf6*<sup>DN</sup> in the *Asap*<sup>K23/B52</sup> combination.

Expression of Arf1<sup>DN</sup> in *Asap*<sup>K23/B52</sup> mutant motor neurons (*Asap*<sup>K23/B52</sup>*; D42-Gal4 > UAS-Arf1*<sup>DN</sup>) or ubiquitously (*Asap*<sup>K23/B52</sup>*; actin5C-Gal4 > UAS-Arf1*<sup>DN</sup>) did not yield viable third instar larvae, so we could not assess epistatic interactions between Arf1 and *Asap* mutants. Although Asap may function through Arf1 in other context, our epistatic interactions suggest that Asap primarily regulates synapse morphology via the Arf6-dependent signaling pathway.

## Constitutively active Arf6 phenocopies the *Asap* mutant and alters active zone organization in a Rab3-dependent manner

Consistent with the hypothesis that Asap functions as a negative regulator of Arf6 activity, motor neuron-specific expression of Arf6<sup>CA</sup> in a wild-type background phenocopied *Asap* mutant (Fig 6G-6G' and 6N-6P). These findings support the idea that excessive Arf6 activation underlies the synaptic defects caused by loss of Asap.

Since Asap loss results in higher Brp density, we further investigated the mechanism underlying Brp reorganization by examining the effect of Arf6 activation and IP3 signaling on Brp distribution. Motor-neuron-specific expression of Arf6<sup>CA</sup> resulted in a marked increase in the number of Brp puncta, accompanied by a reduction in Brp intensity, indicating the formation of numerous but less mature active zones (Fig 7C-7C' and 7G-7H). To test whether this effect of Arf6 depends on Rab3, a well-established regulator of active zone organization [29,41], we analyzed Brp distribution in *rab3* mutant backgrounds. Consistent with prior reports, *rab3* mutants displayed fewer Brp puncta at the larval NMJ (Fig 7A-7B' and 7G-7H). Interestingly, expression of Arf6<sup>CA</sup> in *rab3* mutant background failed to reproduce the Arf6<sup>CA</sup> phenotype. Instead, the Brp organization resembled that of the rab3 mutant, characterized by fewer Brp puncta (Fig 7D-7D' and 7G-7H). Similarly, knockdown of Arf6 in *rab3* mutants showed *rab3* mutant phenotype (Fig 7F-7F' and 7G-7H), suggesting that Arf6-induced active zone formation strictly depends on the functional Rab3.

To further analyze the relationship between Arf6 and Rab3, we examined Rab3 protein levels upon neuronal manipulation of Arf6 activity. Expression of Arf6<sup>CA</sup> or Arf6 RNAi in an otherwise wild-type background resulted in elevated synaptic Rab3 levels, whereas expression of these constructs in the *rab3* mutant background reduced Rab3 signal intensity (Fig 7I-7O). Together, these data indicate that Arf6 acts upstream of or in concert with Rab3 to regulate active-zone organization at the NMJ, likely by modulating membrane or vesicular trafficking events that require Rab3 to correctly distribute active-zone components such as Brp.

We next assessed whether IP3 signaling influences Brp organization in a Rab3-dependent manner by quantifying Brp puncta and Rab3 levels, followed by knockdown of PLCβ, itpr or RyR alone or in *Asap* mutant background. These experiments yielded mixed outcomes. For instance, PLCβ knockdown showed more Brp intensity while knockdown of itpr reduced the Brp intensity without affecting the level of Rab3 (S10 Fig), suggesting that multiple independent and potentially combinatorial processes regulated IP3 signaling may converge on active zone organization. This complexity warrants further investigation to delineate the specific contributions of ER-derived calcium signaling in the regulation of synaptic organization.

## Asap functions through Arf6 to regulate basal synaptic calcium and sustain synaptic activity

Active Arf6 has been implicated in PLC/IP3-mediated calcium release during acrosomal exocytosis [18], and we reasoned that loss of *Asap* could result in elevated levels of Arf6.GTP, leading to increased synaptic calcium release from ER stores. To test if Asap regulates synaptic calcium homeostasis through the Arf6 regulatory pathway, we checked if calcium levels were restored by depleting Arf6 in *Asap* mutants. While motor-neuron specific depletion of Arf6 in otherwise wild-type animals (*D42-Gal4 > UAS-Arf6 RNAi*) did not alter synaptic calcium intensity relative to controls, knockdown of Arf6 in motor

PLOS Genetics

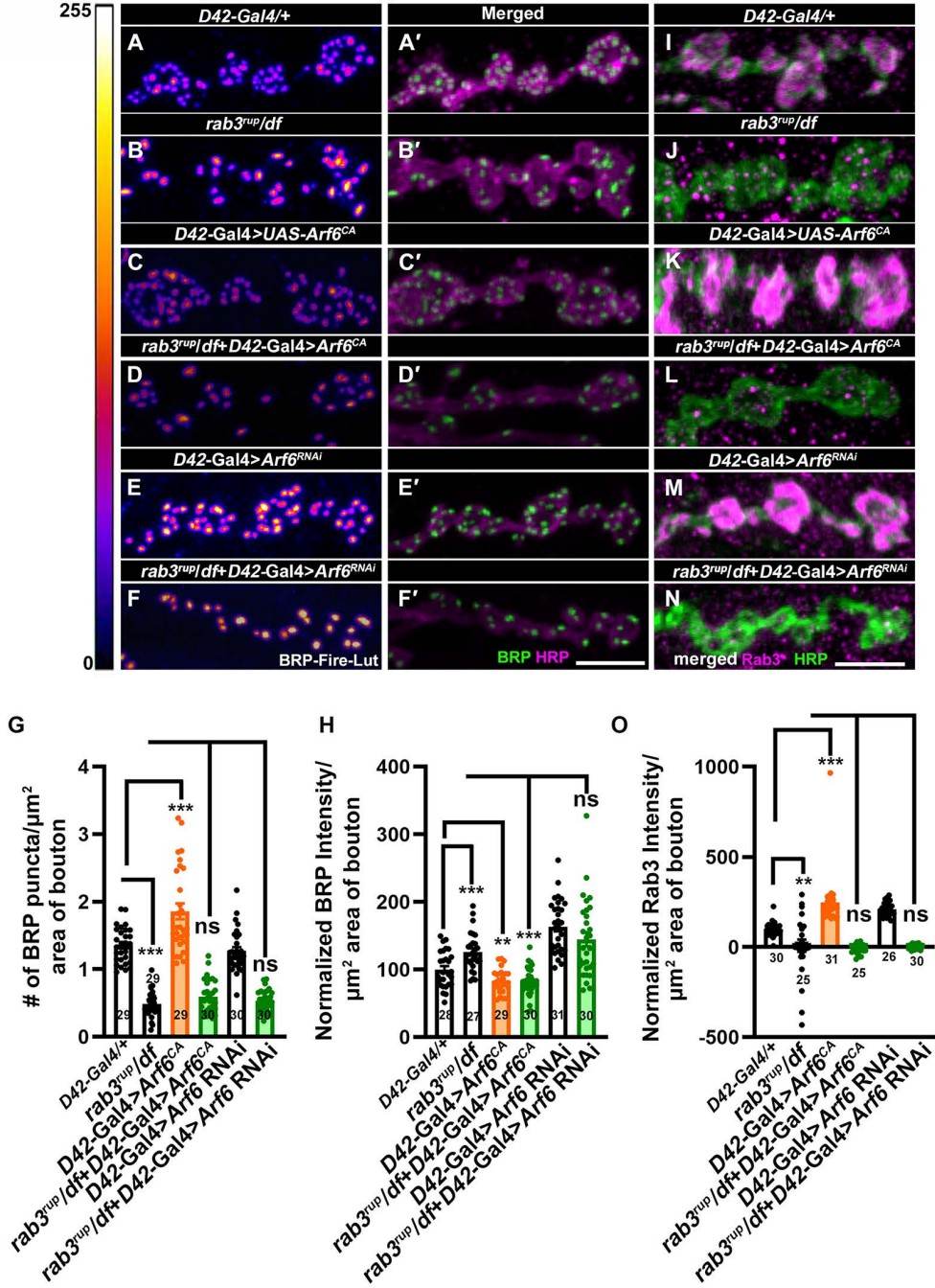

**Fig 7. Arf6-dependent signaling modulates active zone dynamics via Rab3.** (A-F') Confocal images of NMJ synapses at muscle 4 of A2 hemiseg-ment in *D42-Gal4/+* control (A-A'), *rab3^rup/df* (B-B'), *D42-Gal4/UAS-Arf6^CA* (C-C'), *rab3^rup/df; D42-Gal4/UAS-Arf6^CA* (D-D'), *UAS-Arf6 RNAi/+; D42-Gal4/+* (E-E') and *UAS-Arf6 RNAi, df/rab3^rup; D42-Gal4/+* (F-F') double immunolabeled with antibodies against active zones marker Bruchpilot, Brp (green) and HRP (magenta). The scale bar in F' represents 2.5 μm. **(G)** Histogram showing the number of Brp punctae per μm² area of bouton from muscle 4 NMJ at A2 hemisegment in the indicated genotypes. ***p = 0.0001, ***p = 0.0002, n = 29-30 boutons per genotype. All values represent mean ± SEM. **(H)** Histogram showing the levels of Brp per μm² area of bouton from muscle 4 at A2 hemisegment in the indicated genotypes. ***p = 0.001, **p = 0.006, ***p = 0.0001. n = 27-31 boutons per genotype. All values represent mean ± SEM. **(I-N)** Confocal images of NMJ synapses at muscle 4 of A2 hemiseg-ment in *D42-Gal4/+* control **(I)**, *rab3^rup/df* **(J)**, *D42-Gal4/UAS-Arf6^CA* **(K)**, *rab3^rup/df; D42-Gal4/UAS-Arf6^CA* **(L)**, *UAS-Arf6 RNAi/+; D42-Gal4/+* **(M)** and *UAS-Arf6 RNAi, df/rab3^rup; D42-Gal4/+* **(N)** double immunolabeled with antibodies against Rab3 (magenta) and HRP (green). The scale bar in N

represents 2.5 µm. **(O)** Histogram showing the levels of Rab3 per µm$^2$ area of bouton from muscle 4 NMJ at A2 hemisegment in the indicated genotypes. **$p = 0.004$, ***$p = 0.0001$, n = 25-31 boutons per genotype. All values represent mean ± SEM. The values for each quantification are shown in Table G in S1 Text.

neurons of *Asap* mutants (*Asap$^{K23/B52}$; D42-Gal4 > UAS-Arf6 RNAi*) significantly restored synaptic calcium level towards those observed in controls (Fig 8A-8I).

Next, to elucidate whether the gain in synaptic transmission in the *Asap* mutant was mediated through Arf6-dependent signaling, we depleted Arf6 by using *Arf6* RNAi or overexpressed *Arf6$^{DN}$* in *Asap$^{B52/K23}$*. Interestingly, neuronally depleting Arf6 or overexpressing *Arf6$^{DN}$* in *Asap* mutants significantly rescued synaptic failure (Fig 8J-8O and 8V). Moreover, the increased miniature frequency and evoked activity phenotypes in *Asap* mutant NMJs were fully reversed by depleting Arf6 or expressing *Arf6$^{DN}$* transgene in *Asap* mutants (Fig 8P-8U and 8W–8Z). These data strongly suggest that the ArfGAP-containing Asap regulates synaptic function through an Arf6-dependent calcium release at the synapses.

### Arf6-induced synaptic overgrowth and activity require PLCβ-IP3-RyR dependent calcium signaling

Since our data indicate that Asap functions through the Arf6-PLCβ-IP3 signaling pathway to regulate NMJ morphogenesis and basal synaptic calcium handling (Fig 9A), we conducted an epistatic analysis by co-expressing Arf6$^{CA}$ with genetic perturbations that suppress IP3-signaling, thereby directly testing the functional relationship between Arf6 activity and IP3-dependent calcium regulation. We first verified the knockdown efficiency of *itpr*-RNAi, *RyR*-RNAi, or *Plc21C*-RNAi. The qPCR analysis revealed a significant reduction in the corresponding mRNA transcript levels, confirming effective gene silencing (S11 Fig). Next, we coexpressed Arf6$^{CA}$ with *IP₃-sponge*, *itpr*-RNAi, *RyR*-RNAi, or *Plc21C*-RNAi in motor neurons. We found that while the average bouton number was not significantly restored to the control level, the average bouton area and inter-bouton diameter were significantly suppressed upon coexpression of Arf6$^{CA}$ with each of these transgenes (Fig 9B-9L). These data further support that Arf6 regulates synaptic morphogenesis through IP3-dependent pathways. Functional analyses revealed that motor neurons expressing Arf6$^{CA}$ showed larger evoked EPSP amplitudes without changes in mEPSP amplitude or the QC (Fig 9M-9Z). However, coexpression of Arf6$^{CA}$ with the IP3-sponge, or simultaneous expression of IP3-sponge in motor neurons and pharmacological inhibition of RyR with dantrolene, significantly restored EPSP amplitude to control levels. These epistatic interactions further confirm that IP3 and RyR-dependent Ca$^{2+}$ release from the ER underlies Arf6$^{CA}$-induced synaptic gain-of-function.

## Discussion

We demonstrate that Asap, a BAR and ArfGAP domain-containing protein, regulates synaptic resting calcium levels via Arf6-dependent PLCβ signaling at nerve terminals, a pathway essential for synaptic plasticity. Specifically, our findings reveal that Asap (a) governs synaptic morphogenesis by modulating microtubule organization, (b) sustains calcium homeostasis by modulating its release from intracellular stores, and (c) controls synaptic structure and resting synaptic calcium levels through its ArfGAP activity on Arf6.

### Microtubule stability and bouton morphogenesis by Asap

Our findings demonstrate that Asap operates in neurons to maintain microtubule stability, and it plays a significant role in synaptic morphogenesis. Cytoskeletal regulatory proteins are crucial in synapse development across mammals and invertebrates. Disruptions in actin or its regulatory proteins have been shown to impair NMJ formation and signaling [42–44]. Similarly, *Drosophila* mutants with defective microtubule organization exhibit abnormal NMJ development and function [45–47]. Consistent with these findings, we observed that the loss of *Asap* leads to fewer Futsch loops and enlarged boutons. Notably, *Asap* mutant synaptic phenotypes closely resemble those of DAAM mutants [47]. DAAM loss in neurons results in fused

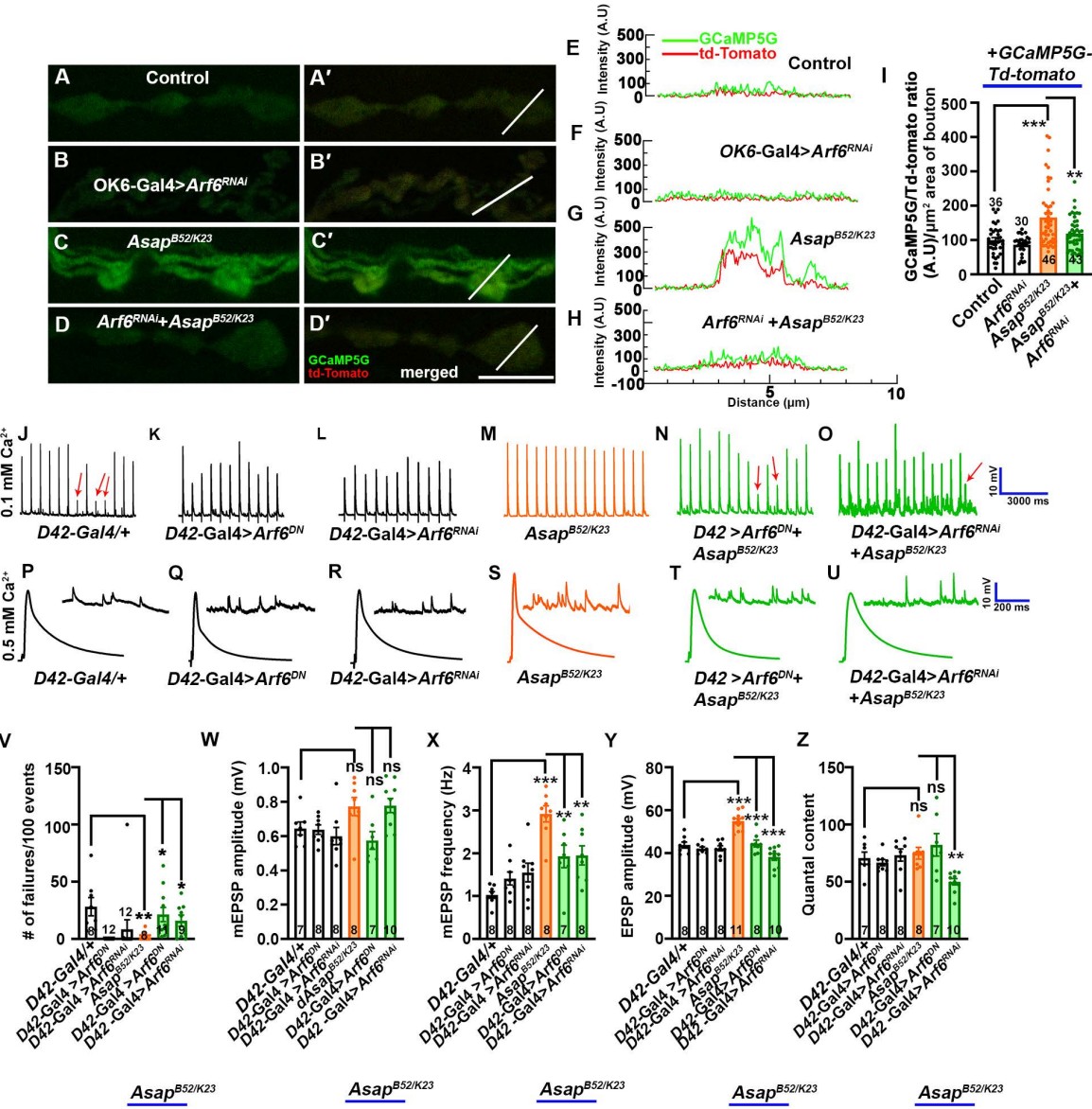

**Fig 8. Active Arf6 maintains synaptic activity by inducing calcium release from ER in *Asap* mutants. (A-D')** Confocal live images of NMJ synapses at muscle 6/7 of A2 hemisegment of (A-A') *OK371-Gal4/ +; UAS-td-Tomato-GCaMP5G/ +*control, (B-B") *OK371-Gal4/UAS-Arf6 RNAi; UAS-td-Tomato-GCaMP5G/ +*, (C-C") *OK6-Gal4, Asap^{K23/B52}; UAS-td-Tomato-GCaMP5G/ +*, (D-D") *UAS-Arf6^{RNAi}, Asap^{B52}/OK6-Gal4, Asap^{K23}; UAS-td-Tomato-GCaMP5G/D42-Gal4* expressing GCaMP5G (green) and tdTomato (red) fluorescent protein. **(E-H)** Intensity plot profile for GCaMP5G (green) and tdTomato (red) across the bouton (shown in A', B', C' and D' as a thin line). The scale bar in D' for (A-D') represents 4 μm. **(I)** Histogram showing the fluorometric ratio of GCaMP5G and tdTomato per μm² bouton area in *OK371-Gal4/ +; UAS-td-Tomato-GCaMP5G/+* (100.0 ± 7.16), *OK371-Gal4/ UAS-Arf6 RNAi; UAS-td-Tomato-GCaMP5G/+* (86.70 ± 4.99), *OK6-Gal4, Asap^{K23/B52}; UAS-td-Tomato-GCaMP5G/+* (165.78 ± 13.11), and *UAS-Arf6^{RNAi}, Asap^{B52}/OK6-Gal4, Asap^{K23}; UAS-td-Tomato-GCaMP5G/D42-Gal4* (119.17 ± 7.54) larvae. **$p = 0.003$ and ***$p = 0.0001$. The statistical analysis was done using one-way ANOVA followed by post-hoc Tukey's multiple-comparison test. n = 20-35 boutons per genotype. All values represent mean ± SEM. **(J-O)** Representative traces of action potential firing at 0.1 mM extracellular Ca²⁺ in **(J)** *D42-Gal4/ +*control, **(K)** *D42-Gal4/UAS-Arf6^{DN}* **(L)** *UAS-Arf6^{RNAi}/ +; D42-Gal4/ +* **(M)** *Asap^{K23/B52}*, **(N)** *Asap^{K23/B52}; D42-Gal4/UAS-Arf6^{DN}* and **(O)** *Arf6 RNAi, Asap^{K23/B52}; D42-Gal4/ +*larvae. The scale bar for EPSPs is x = 3000 ms and y = 10 mV. **(P-U)** Representative traces of mEPSP and EPSP at 0.5 mM extracellular Ca²⁺ in **(P)** *D42-Gal4/ +*control, **(Q)** *D42-Gal4/UAS-Arf6^{DN}*, **(R)** *UAS-Arf6^{RNAi}/ +; D42-Gal4/ +* **(S)** *Asap^{K23/B52}*, **(T)** *Asap^{K23/B52}; D42-Gal4/UAS-Arf6^{DN}* and **(U)** *UAS-Arf6 RNAi, Asap^{K23/B52}; D42-Gal4/ +*larvae. Scale bars for EPSPs (mEPSP) are x = 200 ms (1000 ms) and y = 10 mV (1 mV). **(V)** Histogram showing the percentage of synaptic failures in *D42-Gal4/ +*control, *D42-Gal4/UAS-Arf6^{DN}*, *UAS-Arf6^{RNAi}/ +; D42-Gal4/ +*, *Asap^{K23/B52}*, *Asap^{K23/B52}; D42-Gal4/UAS-Arf6^{DN}* and *UAS-Arf6 RNAi, Asap^{K23/B52}; D42-Gal4/ +*animals. *$p = 0.01$, **$p = 0.005$. n = 8-11 NMJ per genotype. The statistical analysis was done using one-way ANOVA followed by post-hoc Tukey's

multiple-comparison test. All values represent mean±SEM. **(W-Z)** Histogram showing mEPSP amplitude **(W)**, mEPSP frequency **(X)**, EPSP amplitude **(Y)**, and quantal content **(Z)** from muscle 6 of A2 hemisegment in *D42-Gal4/+* control, *D42-Gal4/UAS-Arf6*[DN], *UAS-Arf6*[RNAi]*,/+; D42-Gal4/+, Asap*[K23/B52], *Asap*[K23/B52]*; D42-Gal4/UAS-Arf6*[DN] and *UAS-Arf6 RNAi, Asap*[K23/B52]*; D42-Gal4/+* animals. ns, not significant. ***$p=0.0001$, **$p=0.005$ (for mEPSP frequency), ***$p=0.0001$ (for EPSP amplitude), **$p=0.002$ (for QC). The statistical analysis was done using one-way ANOVA followed by post-hoc Tukey's multiple-comparison test. n=7-11 NMJ per genotype. All values represent mean±SEM. The values for each quantification are shown in Table H in S1 Text.

boutons, increased inter-bouton diameter, and disruptions in the core microtubule cytoskeleton defects remarkably similar to those seen in *Asap* mutants. Our investigation extended beyond the microtubule cytoskeleton to assess whether *Asap* loss affects the actin network. We observed no significant changes in actin levels or the number of moesin puncta at mutant NMJs, suggesting that Asap does not primarily regulate NMJ morphogenesis through actin dynamics.

How might Asap regulate microtubules? A prior genetic study has established that the serine-threonine kinase Shaggy (Sgg)-mediated phosphorylation of Futsch reduces its affinity for microtubules, causing its detachment and leading to presynaptic microtubule destabilization [48]. By contrast, calcineurin, a protein phosphatase that dephosphorylates Futsch under normal Ca²⁺ conditions, counteracts Sgg and promotes microtubule stability [49]. Moreover, resting synaptic calcium levels impact presynaptic microtubule stability [36]. Elevated calcium has been implicated in destabilizing microtubules [50], and more recent data further support the relationship between the posttranslational regulation of microtubules and calcium dependency [51]. Thus, an intricate network of calcium-dependency of Futsch and microtubule stability likely governs NMJ development and morphogenesis [52].

Our observation that *Asap* loss leads to elevated synaptic calcium supports its role in maintaining stable microtubules, potentially through a pathway involving calcium-dependent microtubule-regulatory proteins. Another possibility is that microtubules may play a role in ER channel regulation. Prior studies have shown that microtubule stability influences ER morphology and Ca²⁺ handling [53]. Hence, it is likely that microtubule destabilization in *Asap* mutants may disrupt ER architecture or the spatial positioning of ER Ca²⁺ channels, thereby enhancing their activity. Future experiments assessing microtubule-ER interactions and ER channel localization will be important to directly test this mechanistic link.

Like its mammalian ortholog, Asap also contains a functionally conserved ArfGAP domain [20,54]. Prior data suggest that Asap regulates both Arf1 and Arf6 activities in a tissue-specific manner [22,23]. While the *D42-Gal4>Arf1*[DN]*; Asap*[K23/B52] combination resulted in non-viable larvae, precluding this from further analysis, it does support a crucial involvement of Asap-dependent Arf1 regulation in motor neurons. Several studies implicate Arf6 in regulating multiple processes in neurons, including neurite outgrowth, axon elongation, synapse development, and synaptic vesicle endocytosis [55–58]. Our findings that expressing a dominant-negative Arf6 or depleting Arf6 in *Asap* mutant neurons rescued defects in the microtubule cytoskeleton, bouton morphology, and synaptic transmission at the neuromuscular junction (NMJ), including the functional deficits, strongly support a model that Arf6.GTP is a target for Asap's ArfGAP, and downstream signaling induced by active Arf6 regulates microtubule stability/organization at the *Drosophila* NMJ synapses.

Microtubule disruption reduces the number and density of active zones at the NMJ [45]. In contrast to that expectation, *Asap* mutants have an increase in Brp puncta number and density at the synaptic terminals. Neuronal depletion of Arf6 or expression of a GDP-locked Arf6 in *Asap* mutants rescued not only microtubule defects but also the Brp number and density. Consistent with this, manipulation of Arf6 activity in motor neurons revealed that both constitutively active Arf6 and Arf6 knockdown elevated synaptic Rab3 levels, whereas these effects were attenuated in a *rab3* mutant background. These data collectively suggest that Arf6 functions upstream of or in concert with Rab3 to regulate active-zone organization, likely by modulating vesicular or membrane trafficking events that determine the proper distribution of active-zone components such as Brp [18,29]. While the precise mechanism by which Asap-Arf6 signaling regulates Brp density remains unclear, additional ArfGAP proteins or other Arf family members likely contribute to Brp organization [59]. Further investigation would help to elucidate the broader regulatory network governing Arf6-mediated synaptic organization.

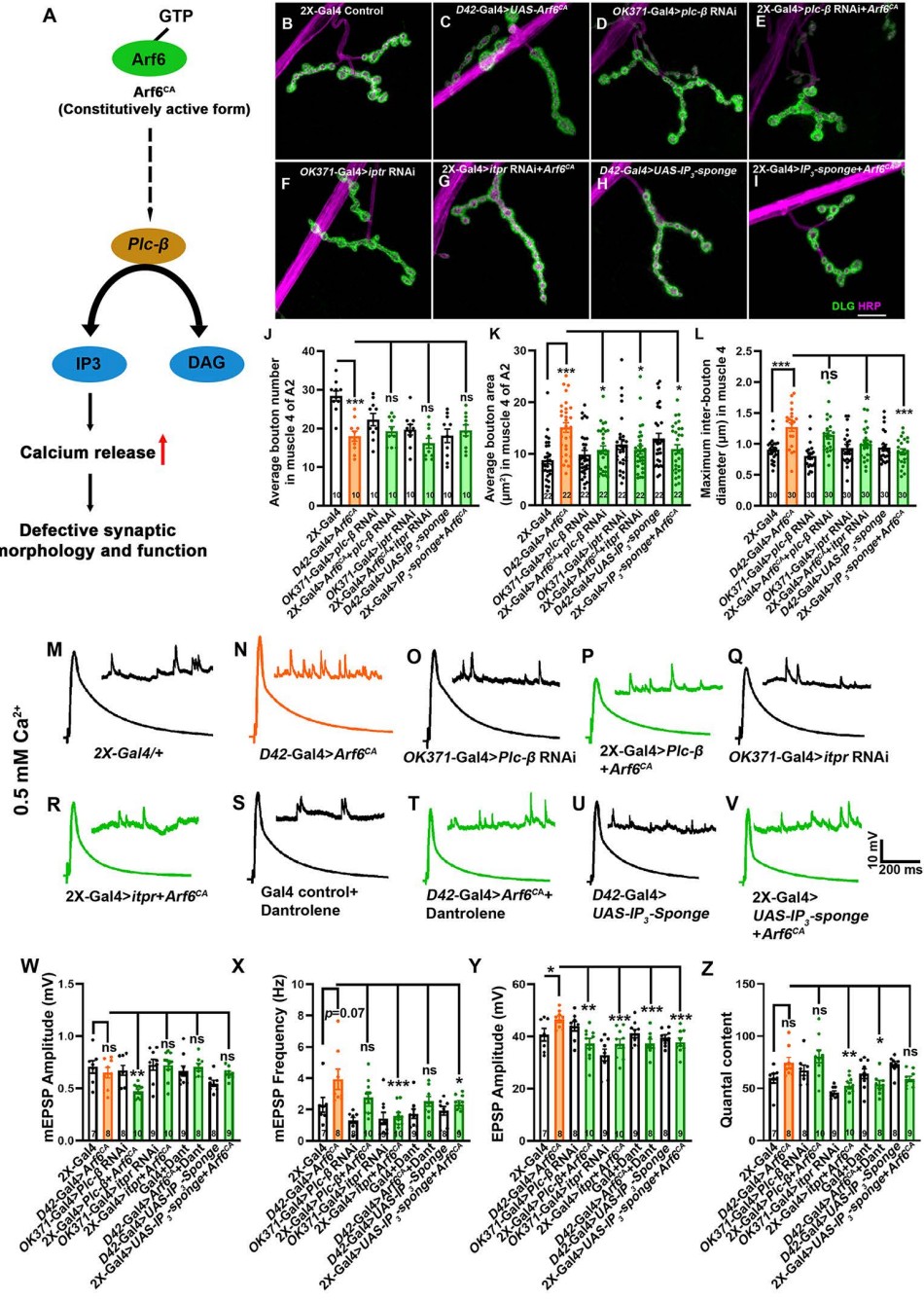

**Fig 9. Constitutively active Arf6 epistatically interacts with IP3 signaling pathway to maintain presynaptic plasticity. (A)** A schematic model illustrating Arf6.GTP-mediated IP3-dependent signaling leading to calcium release. Arf6-GTP activates phospholipase C (PLC), which catalyzes the production of IP3. IP3 binds to IP3 receptors (IP3Rs) on the ER membrane, facilitating calcium release into the cytoplasm, leading to altered synaptic morphogenesis and function. (**B-I**) Confocal images of NMJ synapses at muscle 4 of A2 hemisegment showing synaptic growths in **(B)** *OK371-Gal4/ +; D42-Gal4/ +*, **(C)** *D42-Gal4/ UAS-Arf6^CA*, **(D)** *OK371-Gal4/ UAS-plc-β RNAi*, **(E)** *OK371-Gal4/ UAS-plc-β RNAi; D42-Gal4/ UAS-Arf6^CA*, **(F)** *OK371-Gal4/ UAS-itpr* RNAi, **(G)** *OK371-Gal4/ UAS-itpr RNAi; D42-Gal4/ UAS-Arf6^CA*, **(H)** *OK371-Gal4/ +; UAS-IP_3-sponge/ +*, and **(I)** *OK371-Gal4/ +; UAS-IP_3-sponge, UAS-Arf6^CA/D42-Gal4* double immunolabeled for HRP (magenta) and Dlg (green). Scale bar in I represents 10 μm (B-I). **(J-L)** Histogram showing an average number of boutons **(J)**, average bouton area **(K)**, and maximum interbouton diameter (L) at muscle 4 NMJ of the A2 hemisegment of the indicated genotypes. ns; not significant, ***$p = 0.0001$ (bouton number), ***$p = 0.0001$, *$p = 0.001$ (bouton area), ***$p = 0.001$, *$p = 0.01$ (Inter bouton diameter). The statistical analysis was done using one-way ANOVA followed by post-hoc Tukey's multiple-comparison test. n = 10 NMJ and

22-30 boutons per genotype. All values represent mean ± SEM. (M-V) Representative traces of mEPSP and EPSP in (M) (Control- *OK371-Gal4/ +; D42-Gal4/ +*, (N) *D42-Gal4/UAS-Arf6^{CA}*, (O) *OK371-Gal4/ UAS-plc-β RNAi*, (P) *OK371-Gal4/ UAS-plc-β RNAi; D42-Gal4/ UAS-Arf6^{CA}*, (Q) *OK371-Gal4/ UAS-itpr* RNAi, (R) *OK371-Gal4/ UAS-itpr RNAi; D42-Gal4/ UAS-Arf6^{CA}*, (S) *D42-Gal4* + Dantrolene (20 μM), (T) *D42-Gal4/ UAS-Arf6^{CA}* + Dantrolene (20 μM), (U) *D42-Gal4/UAS-IP_3-sponge* and (V) *OK371-Gal4/ +; UAS-IP_3 sponge, UAS-Arf6^{CA}/D42- Gal4* larvae. Scale bars for EPSPs (mEPSP) are x = 200 ms (1000 ms) and y = 10 mV (1 mV). (W-Z) Histogram showing (W) mEPSP amplitude, (X) mEPSP frequency, (Y) EPSP amplitude and (Z) Quantal content from muscle 6 of A2 hemisegment in *OK371-Gal4/ +; D42-Gal4/ +*, *D42-Gal4/ UAS-Arf6^{CA}*, *OK371-Gal4/ UAS-plc-β RNAi*, *OK371-Gal4/ UAS-plc-β RNAi; D42-Gal4/ UAS-Arf6^{CA}*, *OK371-Gal4/ UAS-itpr* RNAi, *OK371-Gal4/ UAS-itpr RNAi; D42-Gal4/ UAS-Arf6^{CA}*, *D42-Gal4* + Dantrolene (20 μM), *D42-Gal4/ UAS-Arf6^{CA}* + Dantrolene (20 μM), *D42-Gal4/UAS-IP_3-sponge* and *OK371-Gal4/ +; UAS-IP_3 sponge, UAS-Arf6^{CA}/D42- Gal4* animals. **$p = 0.001$; *$p = 0.02$ (mEPSP frequency), *$p = 0.03$, **$p = 0.001$, ***$p = 0.0008$, ***$p = 0.005$ (EPSP amplitude). The statistical analysis was done using one-way ANOVA followed by post-hoc Tukey's multiple-comparison test. n = 7-10 NMJ per genotype. All recordings included in the analysis had an input resistance greater than 5 MΩ. All values represent mean ± SEM. The values for each quantification are shown in Table I in S1 Text.

## Asap loss elevates resting synaptic calcium and transmitter release

The regulation of intracellular calcium homeostasis, particularly through ER-mediated calcium release, is a pivotal determinant of PHP [8]. Under resting conditions, calcium influx through voltage-gated calcium channels (VGCCs) during action potential-driven depolarization facilitates synaptic vesicle fusion and neurotransmitter exocytosis [1,33,60]. However, intracellular calcium mobilization via IP3Rs and RyRs from the endoplasmic reticulum is equally critical in modulating synaptic efficacy, including PHP maintenance at the NMJ [8]. Our data show that Asap mutants exhibit intact PHP responses following acute pharmacological inhibition of postsynaptic glutamate receptors with PhTx, suggesting that the gains in synaptic transmission do not arise from preinduced PHP in *Asap^{K23/B52}* animals.

Instead, the heightened synaptic activity is likely driven by two possible pathways: (a) ER-derived calcium efflux, which induces a more pronounced intracellular calcium elevation than VGCC-dependent influx in PhTx-treated and untreated mutant NMJs, thereby augmenting synaptic vesicle priming and fusion, and (b) an increased number of Brp punctae, which enhances presynaptic release probability. The ER-mediated calcium release likely engages downstream calcium-dependent signaling cascades, including calmodulin and calcium/calmodulin-dependent protein kinase II (CaM-KII), which potentiate neurotransmitter exocytosis by modulating synaptic vesicle dynamics [61]. These findings suggest that calcium mobilization from ER, elevated activity-induced synaptic calcium and Brp upregulation constitute key molecular determinants of the potentiated synaptic response in *Asap* mutants, underscoring the significance of Asap-regulated intracellular calcium signaling in presynaptic homeostatic regulation.

## The interplay of Asap, Arf6, and PLCβ in the regulation of resting synaptic calcium homeostasis

At the NMJ synapses, Asap regulates mini frequency and EPSP amplitude. Loss of *Asap* sustains synaptic vesicle fusion even under low extracellular calcium conditions, reducing synaptic failures (Fig 2). Prior studies in non-neuronal cells have shown that Arf6-GTP regulates PLCβ-IP3-mediated calcium release from intracellular stores (Pelletan et al., 2015), a pathway also critical for synaptic potentiation (He et al., 2000; Yang et al., 2001). Furthermore, ARF6 activation by Gα(q) facilitates the production of IP3, underscoring the role of ARF6 in modulating IP3-mediated signaling [62]. Thus, we postulated that the elevated Arf6-GTP due to the loss of *Asap* might enhance PLCβ-IP3-mediated calcium release from ER stores, potentiating synaptic vesicle fusion at the *Drosophila* NMJ. Indeed, consistent with such a model, depleting PLCβ in *Asap* mutants normalized synaptic transmission, supporting the role of the Asap-Arf6-PLCβ pathway in regulating resting synaptic calcium levels (Figs 5 and 8). This conclusion is further supported by our finding that neuronal expression of Arf6^{DN} or reduction of Arf6 by RNAi restored synaptic calcium to near control levels. Furthermore, epistatic interactions between Arf6^{CA} and itpr or RyR demonstrate that Arf6 modulates both structural and functional plasticity at the NMJ by promoting ER-mediated Ca²⁺ release.

ER and mitochondria are two major sources of calcium handling at the synapses [63,64]. Our data indicate that blockade of either IP3R or RyR can reverse *Asap* mutant phenotypes. This suggests the involvement of both receptors in Asap-Arf6-directed synaptic calcium handling (Fig 4). IP3-regulated calcium release through the IP3R at the

synaptic ER is crucial for the fusion of synaptic vesicles during chronic maintenance of homeostasis at the NMJs [8,65]. Arf6-GTP may contribute to RyR activation by facilitating PLCβ-IP3-mediated calcium release from the ER, which could either directly bind to RyRs or lead to the phosphorylation of enzymes that modulate RyR activity. IP3R-mediated calcium release facilitates the translocation of presynaptic CaMKII to active zones (AZs), thereby enhancing synaptic release probability [66]. Similarly, calcium release via RyRs has been implicated in CaMKII translocation and phosphorylation-driven activation, promoting its recruitment to AZs for efficient neurotransmitter release [66,67]. While RyR functions as a calcium-induced calcium release channel and is modulated by calmodulin, its activity can also be influenced by endogenous kinases and phosphatases [68–72]. However, the precise mechanism by which presynaptic RyR regulation is mediated by Asap remains unclear, despite evidence of PLCβ-IP3-mediated calcium release in *dAsap* mutants (Fig 4). Two potential mechanisms may explain Asap's role in RyR modulation: (1) direct binding to RyRs, altering their calcium release function, or (2) indirect regulation via interactions with upstream signaling components or scaffold proteins. Further investigation is needed to clarify these mechanisms. Additionally, mitochondria contribute to intracellular calcium homeostasis through the Letm1 $Ca^{2+}/H^+$ antiporter, playing a crucial role in calcium dynamics [11,12,73]. Despite this, our findings suggest that Letm1 is not directly involved in Asap-Arf6-PLCβ–mediated synaptic calcium regulation.

We propose a model in which Asap-Arf6-PLCβ signaling is a critical regulator of intracellular calcium homeostasis at *Drosophila* synapses. Five key findings support this model: a) loss of *Asap* results in elevated calcium levels, while chelating intracellular calcium with BAPTA normalized the elevated synaptic activity observed in *Asap* mutant; b) directly inhibiting IP3Rs via an IP3-sponge and RyRs through RNAi restored mEPSP frequency, EPSP amplitude and synaptic calcium to the baseline levels; c) loss of *Asap* exhibit reduced synaptic failure rates, indicative of increased release probability driven by ER-mediated calcium mobilization; d) Arf6 depletion or expression of a dominant-negative Arf6 variant rescued all *Asap* loss associated synaptic defects; and e) epistatic interactions between Arf6$^{CA}$ and IP3-signalling regulators further reinforcing the role of Asap that regulates ER calcium release via Arf6-IP3R/RyR signaling. Beyond neurotransmission, this pathway also influences microtubule-dependent *Drosophila* NMJ development and organization (Fig 10). This study provides the first *in vivo* evidence linking Arf6-dependent signaling to synaptic calcium homeostasis, underscoring a conserved regulatory mechanism governing synaptic function.

## Materials and methods

### *Drosophila* stocks

Flies were raised and maintained under non-crowded conditions at 25°C in standard cornmeal medium. Wild-type $w^{1118}$ were used as controls unless otherwise specified. All the crosses and rescue experiments were conducted at 25°C. The *Drosophila* lines used in this study were: D42-Gal4 (BL-8816), OK6 (BL-64199), OK371-Gal4 (BL-79600), actin5C-Gal4 (BL-25374), and mef2-Gal4 (BL-50742). UAS-Asap-GFP (BL-65849) [74], UAS-Asap-mcherry [23], UAS-tdTomato-P2A-GCaMP5G [36], UAS-IP3-sponge.m30 [75], UAS-Arf6$^{DN}$ (T27N) and UAS-Arf6$^{CA}$ (Q67L) [76], UAS-Arf1$^{DN}$ (T31N) and UAS-Arf1$^{CA}$ (Q71L) [77], Cac$^{sfGFP-N}$ [35], Rab3$^{rup}$ (BL-78045), Df(2R)ED2076 (BL-8909), UAS-Letm1 RNAi (BL-37502), UAS-RyR RNAi (BL-29445), RyR$^{E4340K}$/CyO (BL-5505), UAS-PLCβ RNAi (GD11359), UAS-PLC β RNAi (BL-31269), UAS-itpr RNAi (BL-25937), UAS-itpr RNAi (BL-51686) and itpr$^{90B.0}$/TM6B,Tb$^1$ (BL-30737). The other RNAi lines, UAS-Arf1 RNAi (GD12522 and KK101396) and UAS-Arf6 RNAi (KK108126 and GD13822), were obtained from the Vienna *Drosophila* Resource Centre (VDRC).

### Generation of *Asap* mutant and transgene

To generate the loss-of-function mutants of *Asap*, two sets of gRNAs were designed for the *Asap* genomic region. The first set of gRNAs was designed at the 2$^{nd}$ and 12$^{th}$ exon, while the second set of gRNAs was designed at the 1$^{st}$ and 12$^{th}$ exon using CRISPR Optimal Target Finder online tool. Primers used to clone these gRNAs were: gRNA1FP,

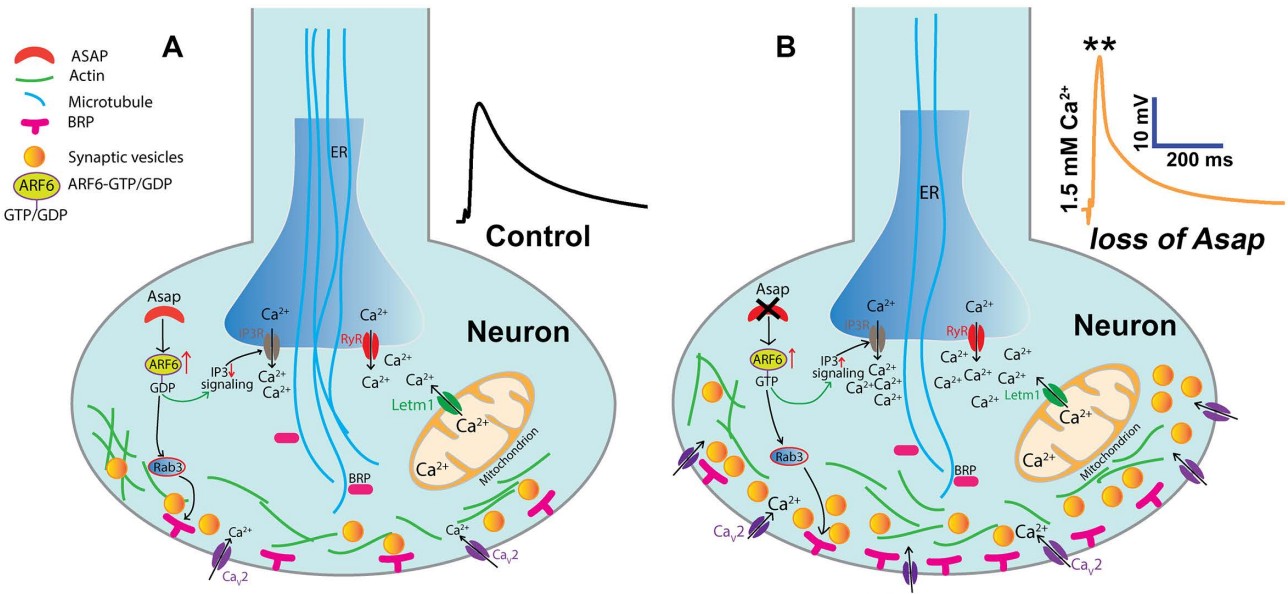

**Fig 10. Proposed model of Asap-Arf6-PLC β signaling in regulating synaptic morphogenesis and calcium dynamics in *Drosophila*.** A model of Asap-mediated Arf6 signaling in *Drosophila* suggests that Asap regulates synaptic stability. Loss of *Asap* activates PLCβ, triggering ER-mediated calcium release, which disrupts microtubule organization at the presynapse and alters bouton morphology. The increased evoked neurotransmission in *Asap* mutants may result from a higher number of active zones and enhanced calcium release from the ER, promoting synaptic vesicle fusion. In wild-type animals, Cav2-type channels regulate synaptic function, whereas in *Asap* mutants, elevated Arf6.GTP likely enhances Rab3 activity, which triggers more Brp punctae assembly and IP$_3$-mediated calcium release, further driving vesicle fusion to maintain synaptic stability. The mitochondrial calcium does not appear to contribute to vesicle fusion, highlighting ER-mediated calcium release as the key driver of increased neurotransmission in *Asap* mutants. The models in panel (A-B) are created using Adobe Illustrator.

5′- TATATAGGAAAGATATCCGGGGTGAACTTCGAGATTGGAGTTCGACCGGGGGTTTTAGAGCTAGAAATAGCAAG-3′, and gRNA1RP, 5′-ATTTTAACTTGCTATTTCTAGCTCTAAAACCAATAATGCCCTCCATCCAGCGACGTTAAATT GAAAATAGGTC-3′; gRNA2FP, 5′-TATATAGGAAAGATATCCGGGGTGAACTTCGCCCGACTGCCGCCACACGATGTTTTA GAGCTAGAAATAGCAAG-3′, and gRNA2RP: 5′-ATTTTAACTTGCTATTTCTAGCTCTAAAACCGCAATCGTAGAGAGCTC GGCGACGTTAAATTGAAAATAGGTC-3′. The gRNAs were cloned into a dual gRNA pCFD4 vector having a BbsI restriction site using Gibson Assembly Kit (New England Biolabs (UK) Ltd) following the manufacturer's guidelines. The pCFD4 vector containing *Asap* gRNAs was injected into *Drosophila* embryos to generate the transgene. Next, the transgenic flies containing the *Asap* gRNAs were crossed with *Nanos-cas9* (BL-54591) to create the deletion of *Asap* gene in the germline cells. Following standard genetics, lines were established in F2 generation, and *Asap* deletion was screened by PCR using primers, 5′-TTGGCTAAGTTTGGGAATGC-3′ and 5′-GAGTTTAAAAGACGCATACACACA-3′. Two null mutants, *Asap*^K23^ (4190 bp deletion) and *Asap*^B52^ (4275 bp deletion), were obtained. To generate *Asap* transgene, a full-length *Asap* ORF was amplified from cDNA and cloned in Gal4-based expression vector pUASp at the blunt-ended NotI restriction site. The pUASp vector containing the *Asap* ORF was injected into *Drosophila* embryos to generate the transgene.

## Semiquantitative RT-PCR

The expression of *Asap* was analyzed by semiquantitative RT-PCR. In brief, total RNA was isolated from larval fillets using TRIzol reagent (Invitrogen, Thermo Fisher Scientific, Waltham, MA, USA). Reverse transcription was performed on 1 μg total RNA using Superscript II Reverse Transcriptase (Invitrogen, Thermo Fisher Scientific, Waltham, MA, USA) using an oligo-dT primer to make cDNA. The resulting cDNA was used for PCR to analyze the level of *Asap* transcript using

primers: 5'-CAAAGATTCCTCCACCAGGA-3' and 5'-GATCTAGACCGCCACCAGAG-3'. *rp49* was used as an internal control for PCR. The primer sequence used for the amplification of *rp49* was 5'-AGATCGTGAAGAAGCGCACC-3' and 5'-CGATCCGTAACCGATGTTGG-3'.

## Quantitative RT-PCR (qRT-PCR)

The Qiagen RNA extraction kit was used to isolate total RNA from larval fillets, following the manufacturer's instructions (Invitrogen, MA, USA). The Takara PrimeScript 1st Strand cDNA Synthesis Kit was used to synthesize first-strand cDNA (Takara, 6110A). Quantitative RT-PCR reactions were set up using iTaq Universal SYBR Green Supermix (#1725124, Bio-Rad) in the qTOWER³ (Analytik Jena, Jena, Germany) qPCR machine. The primers were designed using the IDT Primer Quest tool (https://www.idtdna.com) (Table T in S1 Text). *rp49* was used as an internal control. Three independent RT-qPCR runs were performed. The fold change was calculated using $2^{-\Delta(\Delta Ct)}$ [78] method.

## Generation and affinity purification of antibody

To generate antibodies against Asap, the N-terminal 412 amino acids of Asap (amino acids 1–412) were cloned in the pET-28a (+) bacterial expression vector. The His-tagged fusion protein was expressed in BL21 codon+ cells, purified from inclusion bodies, and injected into rabbits (Deshpande Laboratories, Bhopal, India). For affinity purification of the anti-Asap antibodies, 200 µg of the His-tagged Asap was resolved on 12% SDS-PAGE, transferred to PVDF membrane, and then incubated with 1.0 mL of anti-Asap serum diluted 50 folds in 1X PBS (pH 7.4) overnight at 4°C. The membrane was rinsed five times with 1X PBS (pH 7.4). The bound antibody was eluted by incubating the membrane with 500 µl of elution buffer (100 mM glycine; pH 2.5) for 20 minutes at room temperature. The pH was neutralized immediately by adding 0.1 volume of 1.0 M Tris base, pH 10. The eluted antibody was kept at 4°C for immediate use or stored at -20°C.

## Western blot analysis

Adult fly heads were homogenized in lysis buffer (50 mM Tris-HCl, pH 6.8, 25 mM KCl, 2.0 mM EDTA, 0.3 M sucrose, and 2% SDS) at 75°C in a water bath. The protein concentration was quantified using bicinchoninic acid (BCA) Protein assay (Sigma-Aldrich, St. Louis, Missouri, USA). Then, 60 µg of the total protein of each genotype was separated on 8% SDS-PAGE and transferred to the PVDF membrane (Amersham, GE Healthcare Life Sciences, Illinois, USA). The membrane was blocked in 5% skimmed milk (HIMEDIA, Maharashtra, India) in 1X Tris-buffered saline (TBS) with 0.2% Tween-20 (0.2% TBST) for 1 hour at room temperature and then incubated overnight with primary antibody. After washing with 0.2% TBST, the membrane was incubated with HRP-conjugated secondary antibody (1:10000) for 1 hour at room temperature. The primary antibody used was rabbit anti-Asap (this study, 1:1000) and mouse anti-Ran (1:3000, BD Biosciences, New Jersey, USA). Signals were detected using the Odyssey imaging system (LI-COR Biosciences, Lincoln, USA).

## Immunohistochemistry

Wandering third instar larvae were dissected in ice-cold $Ca^{2+}$-free HL3 saline and fixed in either 4% PFA for 30 min or Bouin's fixative (Sigma Aldrich, St. Louis, Missouri, USA) for 10 min [21,79]. The fillets were incubated with primary antibodies overnight at 4°C, followed by secondary antibodies at room temperature for 90 minutes. Finally, larval fillets were washed with 0.2% PBST and mounted with Fluoromount aqueous mounting medium (Sigma-Aldrich, St. Louis, Missouri, USA) on a glass slide. The monoclonal antibody anti-CSP (ab49, 1:100), anti-Futsch (22C10, 1:100), anti-Dlg1 (4F3, 1:30) and anti-Bruchpilot (nc82, 1:50) were obtained from Developmental Studies Hybridoma Bank (DSHB, University of Iowa, USA). Anti-GluRIII (1:100) [80], anti-Rab3A (1:500) [29] antibodies were a kind gift from Aaron DiAntonio (Washington University, St. Louis, USA). The FluoTag-X4 anti-GFP nanobodies (1:200) were obtained from NanoTag Biotechnologies (Göttingen, Germany). The fluorophore-conjugated secondary antibody Alexa Fluor 488 or Alexa Fluor 568 (Invitrogen,

Thermo Fisher Scientific, Waltham, MA, USA) was used at 1:800 dilution. Alexa Fluor 488 or Rhodamine conjugated anti-HRP (Jackson ImmunoResearch, Baltimore, PA, USA) were used at 1:800 dilution. Images were captured with a laser scanning confocal microscope (Olympus FV3000 or LSM780, Carl Zeiss) using 40x/1.3 NA or 60x/1.42 NA objectives. The images were processed using Image J (ImageJ, NIH, USA) or Adobe Photoshop software (Adobe Inc., USA).

## Morphometric analysis and quantification

Muscle 4 or muscle 6/7 of A2 hemisegment was used for NMJ morphology quantification. The CSP-positive structures were counted to quantify the total bouton number. For bouton area quantification, CSP-marked boundaries were used to define one bouton. The average of the five biggest boutons from each NMJ was used to quantify bouton size and area. 'Fused bouton' phenotype was analyzed using the maximum inter-bouton diameter, measured on three neighboring inter-bouton regions in each sample, and the average of these three regions was used as one data point for quantification [47]. For Futsch loop quantification, the third instar larval fillets were immunolabelled with HRP and 22C10 antibodies. The NMJs of muscle four were imaged with 40X objective, and futsch-loop structures that localized with HRP were counted. The total number of Futsch loops was divided by the total number of boutons to calculate the percentage of boutons with loops [46]. To quantify Brp intensity, two independent experimental sets were analyzed: one comprising unchallenged and PhTx-challenged control fillets, and the other comprising unchallenged and challenged *Asap* mutant fillets. Comparisons were made within each genotype rather than between genotypes. To quantify the GluRIII clusters at the larval NMJ, confocal images were analyzed by identifying GluRIII puncta apposed to active zone puncta. A binary mask was generated in ImageJ to delineate the boundaries of each GluRIII cluster, and the corresponding area was measured in square micrometers ($\mu m^2$).

For multiple comparisons, one-way ANOVA followed by Post-hoc Tukey's test was used. GraphPad Prism 10 was used to plot the graph. Error bars in bar graphs represent the standard error of the mean (SEM).

## Calcium imaging

Third-instar *Drosophila* larval fillets were dissected in HL3 saline containing (in mM): 70 NaCl, 5 KCl, 0.5 CaCl$_2$, 10 MgCl$_2$, 10 NaHCO$_3$, 5 trehalose, 115 sucrose, and 5 HEPES. Following dissection, the fillets were equilibrated at room temperature in HL3 containing 0.5 mM Ca$^{2+}$ for at least 5–7 minutes before imaging. To prevent muscle contractions during the experiment, the larval fillets were imaged in ice-cold saline, and the ventral nerve cords (VNCs) were severed from the dorsal brain lobes to eliminate peristaltic activity as described previously [36]. Type 1b boutons on muscle M6/7 of abdominal segment A2 were visualized using a glass bottom chamber in a 63 × inverted oil-immersion objective on a Zeiss 700 microscope. Initial fluorescence signals were seen under the microscope for tdTomato and GCaMP5G. The bath solution was then replaced with ice-cold HL3 containing 0.5 mM CaCl$_2$, and fluorescence images were acquired two minutes after solution exchange under identical imaging conditions. Image analysis was performed using Image J software. Background fluorescence was determined by selecting regions devoid of axon terminals but adjacent to the region of interest (ROI). The average pixel intensity from background regions was subtracted from the ROI for each fluorescence channel to ensure accurate quantification. For the stimulation experiments, third-instar larvae were dissected in a glass-bottomed chamber in HL3 saline, as noted earlier. For the stimulation experiments, third-instar larvae were dissected in a glass-bottomed chamber in HL3 saline, as noted earlier. For KCl stimulation, HL3 saline was replaced with ice-cold HL3 containing 90 mM KCl. The images were captured at the same laser intensity for unstimulated and 90 mM KCl-stimulated control and mutant animals.

## Electrophysiology and pharmacology

Intracellular recordings were performed from muscle 6 of A2 hemisegment as described previously [38,81]. Briefly, wandering instar larvae were dissected in HL3 saline containing 1.5 mM Ca$^{2+}$ at room temperature. Miniature excitatory postsynaptic potentials (mEPSPs) were recorded for 60 seconds. Evoked excitatory postsynaptic potentials

(EPSP) were recorded from the muscle at 1 Hz nerve stimulation. The EPSP was recorded by delivering a depolarizing pulse, and the signal was amplified using Axoclamp 900A. The data was digitized using Digidata 1440A and acquired using pClamp7 software (Axon Instruments, Molecular Devices, USA). Intracellular recording microelectrodes having resistance between 20–30 MΩ, filled with the 3M KCl, were used for all recordings. Data from muscles showing resting membrane potential ranging from -60 mV to -70 mV and muscle resistanve resistance greater than 5MΩ were used for analysis. The quantal content was calculated as the ratio of the average EPSP amplitude to the average mEPSP amplitude for each NMJ.

As indicated in the figures, the pharmacological agents were mixed in HL3 saline at the final concentrations. The drugs used in the study were Philanthotoxin-433 (PhTx) (Sigma Aldrich), BAPTA-AM (Sigma Aldrich), EDTA-AM (AAT Bioquest) and Dantrolene (Tocris). Failure analysis was performed at 0.1 mM $Ca^{2+}$ containing HL3. One hundred trials (stimulations) were performed at each NMJ in all the genotypes. The failure rate percentage was obtained by dividing the total number of failures by the total number of trials. Paired-pulse ratios were calculated as the EPSP amplitude of the second response divided by the first response (EPSP2/EPSP1). The data were analyzed using Mini Analysis (Synaptosoft) and Clampfit software (Molecular Devices, USA).

## Supporting information

**S1 Fig. The *Asap* mutants are protein null and alter NMJ morphology. (A)** Semi-quantitative RT-PCR showing *Asap* transcript level in *w^1118* controls, homozygous *Asap^K23*, homozygous *Asap^B52*, and heteroallelic *Asap^K23/B52* mutant animals. *rp49* transcript levels were used as an internal loading control. **(B)** Western blot showing protein levels of Asap in *w^1118* controls, homozygous *Asap^K23*, homozygous *Asap^B52*, and heteroallelic *Asap^K23/B52* mutant animals. Ran protein levels were used as an internal loading control. **(C-F′)** Confocal images of NMJ synapses at muscle 4 of A2 hemisegment in (C-C′) *w^1118* control, (D-D′) homozygous *Asap^K23*, (E-E′) homozygous *Asap^B52*, and (F-F′) heteroallelic *Asap^K23/B52* mutant animals double immunolabeled for HRP (green) and CSP (magenta). The scale bar in F′ for (C-F′) represents 10 μm. **(G-J)** Histogram showing an average number of boutons (G), average bouton area (H), average bouton size (I), and maximum inter-bouton diameter (J) at muscle 4 NMJ of the A2 hemisegment of the indicated genotypes. **$p = 0.001$, ***$p = 0.0001$; ns, not significant. n = 14–16 NMJ per genotype. The statistical analysis was done using one-way ANOVA followed by post-hoc Tukey's multiple-comparison test. All values represent mean ± SEM. The values for each quantification are shown in Table J in S1 Text.
(TIF)

**S2 Fig. Asap antibodies detect Asap epitopes at the NMJ, and overexpressed protein gets targeted to the NMJ synapses. (A)** Domain organization of Asap. The N-terminal 1–412 aa was used to generate antibodies against Asap. **(B-C")** Confocal images of NMJ synapses at muscle 4 of A2 hemisegment in the (B-B") wild-type (*w^1118*) and (C-C") *Asap^K23/B52* heteroallelic mutant, immunolabeled with HRP (magenta) and dAsap antibody (1–412 aa) (green). The scale bar in C" for (B-C") is 20 μm. **(D-E")** (D-D") Represents confocal images of NMJ synapses at muscle 4 of the A2 hemisegment of wild-type, double immunostained with the Asap antibody (magenta) and HRP (green), highlighting its distribution at the NMJ. In contrast, (E-E") represents the immunostaining of the wild-type NMJ using Asap antibody, which was preabsorbed with pure Asap protein (1–412 a.a.), resulting in no detectable staining at the NMJ. The scale bar in E" applies to panels D-E" and represents 2.5 μm. **(F)** Western blotting showing protein levels of Asap in different tissues in wild-type and *Asap^K23/B52* animals. Ran protein levels were used as an internal loading control. **(G-H")** Confocal images of NMJ synapses at muscle 6–7 of the A2 hemisegment of (G-G") *UAS-Asap-mcherry/+* and (H-H") *elav^c155-Gal4 > UAS-Asap-mcherry* immunostained with HRP (green) and mcherry (magenta). The scale bar in H" for (G-H") is 20 μm.
(TIF)

**S3 Fig. Control expression of *Asap* transgene or the Gal4 driver alone shows no rescue of *Asap* mutant phenotype. (A-C)** Confocal images of NMJ synapses at muscle 4 of A2 hemisegment showing synaptic growths in (A) *D42-Gal4/* +control, (B) *Asap^{B52/K23}; D42-Gal4/*+ and (C) *Asap^{B52/K23}; UAS-Asap^{FL}/* +double immunolabeled for CSP (magenta) and HRP (green). Scale bar represents 10 μm (A-C). **(D-F)** Histogram showing an average number of boutons (D), average bouton area (E), and maximum interbouton diameter (F) at the muscle 4 NMJ of the A2 hemisegment of the indicated genotypes. ns; not significant, ***$p = 0.0002$ (bouton number), ***$p = 0.0001$ (bouton area), ***$p = 0.0001$ (inter bouton diameter). The statistical analysis was done using one-way ANOVA followed by post-hoc Tukey's multiple-comparison test. n = 8 NMJ per genotype. **(G-I)** Representative traces of mEPSP and EPSP in (G) *D42-Gal4/* +control, (H) *Asap^{B52/K23}; D42-Gal4/*+ and (I) *Asap^{B52/K23}; UAS-Asap^{FL}/* +larvae. Scale bars for EPSPs (mEPSP) are x = 200 ms (1000 ms) and y = 10 mV (1 mV). **(J-M)** Histogram showing mEPSP amplitude (J), mEPSP frequency (K), EPSP amplitude (L) and Quantal content (M) from muscle 6 of A2 hemisegment in the indicated genotypes. ***$p = 0.0001$, ***$p = 0.0006$ (mEPSP frequency), ***$p = 0.0005$, *** $p = 0.0001$ (EPSP amplitude). The statistical analysis was performed using Student's t-test for pairwise comparisons. n = 7–10 NMJ per genotype. All recordings included in the analysis have an input resistance greater than 5 MΩ. All values represent mean ± SEM. The values for each quantification are shown in Table K in S1 Text.
(TIF)

**S4 Fig. Microfilament-based cytoskeleton is normal in *Asap* mutants. (A-C')** Representative confocal images of NMJ synapses at muscle 4 of A2 hemisegment showing the microtubule loops in (A-A') control, (B-B') *Asap^{K23/B52}*, and (C-C') *Asap^{K23/B52}; D42-Gal4/UAS-Asap* double immunolabeled with ace-tubulin (magenta) and HRP (green). Scale bar in C' (for A-C') represents 10 μm. **(D)** Histogram showing the percentage of microtubule-positive loops from muscle 4 NMJ at A2 hemisegment in *D42-Gal4/* +control (52.14 ± 2.84%), *Asap^{K23/B52}* (19.47 ± 2.23%), *Asap^{K23/B52}; D42-Gal4/ UAS-Asap* (48.95 ± 3.73%) animals. The error bar represents the standard error of the mean (mean ± SEM); the statistical analysis was done using one-way ANOVA followed by post-hoc Tukey's test. ***$p = 0.0001$; ns, not significant. **(E)** Histogram showing the ace-tubulin intensity normalized with HRP from muscle 4 NMJ at A2 hemisegment in *D42-Gal4/* +control, *Asap^{K23/B52}*, *Asap^{K23/B52}; D42-Gal4/UAS-Asap* animals. The error bar represents the standard error of the mean (SEM); the statistical analysis was done using one-way ANOVA followed by post-hoc Tukey's test. ns, not significant. **(F)** Western blots showing ace-Tubulin and α-Tubulin protein levels in the indicated genotypes. Ran protein levels were used as an internal loading control. **(G)** Histogram showing the quantification percentage of ace-tubulin level in control, *Asap^{K23/B52}*, *Asap^{K23/B52}; D42-Gal4/UAS-Asap* animals. The error bar represents the standard error of the mean (SEM); the statistical analysis was done using one-way ANOVA followed by post-hoc Tukey's test. ns, not significant. **(H)** Histogram showing the quantification percentage of α-tubulin level in control (1.00 ± 0.00), *Asap^{K23/B52}* (1.16 ± 0.08), *Asap^{K23/B52}; D42-Gal4/UAS-Asap* (1.10 ± 0.12) animals. The error bar represents the standard error of the mean (SEM); the statistical analysis was done using one-way ANOVA followed by post-hoc Tukey's test. ns, not significant. **(I-J')** Representative confocal images of NMJ synapses at muscle 4 of A2 hemisegment showing the number of moesinGFP punctae in (I-I') control: *D42-Gal4/UAS-moesinGFP*, (J-J') *Asap^{K23/B52}; D42-Gal4/UAS-moesinGFP* double immunolabeled with HRP (magenta) and anti-GFP (green). Scale bar in J' for (I-J') represents 10 μm. **(K-L')** Representative confocal images of NMJ synapses at muscle 4 of A2 hemisegment showing the number of ActinGFP punctae in (K-K') control: *D42-Gal4/UAS-ActinGFP*, (L-L') *Asap^{K23/B52}; D42-Gal4/UAS-ActinGFP* double immunolabeled with HRP (magenta) and anti-GFP (green). Scale bar in L' for (K-L') represents 10 μm. **(M)** Histogram showing the number of moesinGFP punctae per μm² area of NMJ from muscle 4 NMJ at A2 hemisegment in *D42-Gal4/UAS-moesinGFP*, and *Asap^{K23/B52}; D42-Gal4/UAS-moesinGFP* animals. The error bar represents the standard error of the mean (SEM); the statistical analysis was done using one-way ANOVA followed by post-hoc Tukey's test. ns, not significant. **(N)** Histogram showing the number of ActinGFP punctae per μm² area of NMJ from muscle 4 NMJ at A2 hemisegment in *D42-Gal4/UAS-ActinGFP*, and *Asap^{K23/B52}; D42-Gal4/UAS-ActinGFP* animals. The error bar represents the standard error

of the mean (SEM); the statistical analysis was done using one-way ANOVA followed by post-hoc Tukey's test. ns, not significant. The values for each quantification are shown in Table L in S1 Text.
(TIF)

**S5 Fig. Loss of *Asap* triggers a high probability of synaptic release at the terminals. (A)** Representative paired-pulse traces of action potential firing at 0.1 mM extracellular $Ca^{2+}$ in *D42-Gal4/ +* control, *Asap$^{K23/B52}$*, *Asap$^{B52/K23}$; UAS-Asap$^{FL}$/D42-Gal4* and *Asap$^{B52/K23}$; D42-Gal4/Arf6$^{DN}$* larvae. The scale bar for EPSPs is x = 200 ms and y = 10 mV. **(B)** Histogram showing paired-pulse ratio (EPSP2/EPSP1) in control, *Asap$^{B52/K23}$*, *Asap$^{B52/K23}$; UAS-Asap$^{FL}$/D42-Gal4,* and *Asap$^{B52/K23}$; D42-Gal4/UAS-Arf6$^{DN}$* animals. ns, not significant, *$p = 0.045$. n = 6–11 NMJ per genotype. The statistical analysis was done using one-way ANOVA followed by post-hoc Tukey's multiple-comparison test. Values represent mean ± SEM. **(C)** Representative paired-pulse traces of action potential firing at 0.4 mM extracellular $Ca^{2+}$ in control, *Asap$^{K23/B52}$*, *Asap$^{B52/K23}$; UAS-Asap$^{FL}$/D42-Gal4*, and *Asap$^{B52/K23}$; D42-Gal4/Arf6$^{DN}$* larvae. The scale bar for EPSPs is x = 200 ms and y = 10 mV. **(D)** Histogram showing paired-pulse ratio (EPSP2/EPSP1) in control, *Asap$^{B52/K23}$*, *Asap$^{B52/K23}$; UAS-Asap$^{FL}$/D42-Gal4,* and *Asap$^{B52/K23}$; D42-Gal4/UAS-Arf6$^{DN}$* animals. ns, not significant, **$p = 0.004$. n = 10–11 NMJ per genotype. The statistical analysis was done using one-way ANOVA followed by post-hoc Tukey's multiple-comparison test. All values represent mean ± SEM. **(E)** Representative paired-pulse traces of action potential firing at 1.5 mM extracellular $Ca^{2+}$ in control, *Asap$^{K23/B52}$*, *Asap$^{B52/K23}$; UAS-Asap$^{FL}$/D42-Gal4*, and *Asap$^{B52/K23}$; D42-Gal4/Arf6$^{DN}$* larvae. The scale bar for EPSPs is x = 200 ms and y = 10 mV. **(F)** Histogram showing paired-pulse ratio (EPSP2/EPSP1) in control, *Asap$^{B52/K23}$*, *Asap$^{B52/K23}$; UAS-Asap$^{FL}$/D42-Gal4,* and *Asap$^{B52/K23}$; D42-Gal4/UAS-Arf6$^{DN}$* animals. ns, not significant. n = 9–10 NMJ per genotype. The statistical analysis was done using one-way ANOVA followed by post-hoc Tukey's multiple-comparison test. All values represent mean ± SEM. **(G)** Representative traces of EPSPs and mEPSPs in control, *Asap$^{B52/K23}$*, *Asap$^{B52/K23}$; UAS-Asap$^{FL}$/D42-Gal4,* and *Asap$^{B52/K23}$; D42-Gal4/UAS-Arf6$^{DN}$* larvae in 0.1 mM $Ca^{2+}$ concentration. The scale bar for EPSPs is x = 200 ms and y = 10 mV. **(H)** Histogram showing EPSP amplitude from muscle 6 of A2 hemisegment in control, *Asap$^{B52/K23}$*, *Asap$^{B52/K23}$; UAS-Asap$^{FL}$/D42-Gal4,* and *Asap$^{B52/K23}$; D42-Gal4/UAS-Arf6$^{DN}$* animals. ns, not significant, **$p = 0.006$. n = 9–14 NMJ per genotype. The statistical analysis was done using one-way ANOVA followed by post-hoc Tukey's multiple-comparison test. All values represent mean ± SEM. The values for each quantification are shown in Table M in S1 Text.
(TIF)

**S6 Fig. High KCl stimulation evokes a heightened calcium response in *Asap* mutants. (A, A' and C, C')** Confocal live images of NMJ synapses at muscle 6/7 of the A2 hemisegment of (A-A') control-unstimulated and (C-C') control-stimulated with 90 mM KCl, expressing GCaMP5G (green) and td-Tomato (magenta) fluorescent protein. The scale bar in C' for (A- A', C- C') represents 4 µm. **(B, D)** Intensity plot profile for GCaMP5G (green) and td-Tomato (magenta) across the bouton (shown in A' and C' and thin line). **(E)** Histogram showing the fluorometric ratio of GCaMP5G and td-Tomato per µm² bouton area in *OK-371-Gal4/ +; UAS-GCaMP5G-td-Tomato/ +* control, unstimulated (100.0 ± 4.39) and *OK-371-Gal4/ +; UAS-GCaMP5G-td-Tomato/ +* control, stimulated (130.79 ± 16.73) larvae. $p = 0.08$. The statistical analysis was performed using Student's t-test for pairwise comparisons. n = 26 boutons per genotype. All values represent mean ± SEM. **(F, F' and H-H')** Confocal live images of NMJ synapses at muscle 6/7 of A2 hemisegment of (F-F') *Asap$^{B52/K23}$*-unstimulated and (H-H') *Asap$^{B52/K23}$*-stimulated with 90 mM KCl expressing GCaMP5G (green) and td-Tomato (magenta) fluorescent protein. The scale bar in H' for (F- F', H- H',) represents 4 µm. **(G, I)** Intensity plot profile for GCaMP5G (green) and td-Tomato (magenta) across the bouton (shown in F' and H' and thin line). **(J)** Histogram showing the fluorometric ratio of GCaMP5G and td-Tomato per µm² bouton area in *OK371-Gal4, Asap$^{B52/K23}$; UAS-GCaMP5G-td-Tomato/ +,* unstimulated (100.0 ± 6.00) and *OK371-Gal4, Asap$^{B52/K23}$; UAS-GCaMP5G-td-Tomato/ +,* stimulated (204.7 ± 33.81) larvae. $p = 0.001$. The statistical analysis was performed using Student's t-test for pairwise comparisons. n = 22–26 boutons per genotype. All values represent mean ± SEM. The values for each quantification are shown in Table N in S1 Text.
(TIF)

**S7 Fig. Depleting IP$_3$ signaling components restores heightened calcium levels in *Asap* mutants. (A-H')** Confocal live images of NMJ synapses at muscle 6/7 of A2 hemisegment of (A-A') *OK-371-Gal4/ +; UAS-td-Tomato-GCaMP5G/ +*, (B-B') *OK371-Gal4, Asap$^{B52/K23}$; UAS-td-Tomato-GCaMP5G/ +*, (C-C') *OK371-Gal4/ +; UAS-td-Tomato-GCaMP5G/ UAS-itpr RNAi*, (D-D') *OK371-Gal4, Asap$^{B52}$/OK371-Gal4, Asap$^{K23}$; UAS-td-Tomato-GCaMP5G/UAS-itpr RNAi*, (E-E') *OK371-Gal4/ +; UAS-td-Tomato-GCaMP5G/ UAS-RyR RNAi*, (F-F') *OK371-Gal4, Asap$^{B52}$/OK371-Gal4, Asap$^{K23}$; UAS-td-Tomato-GCaMP5G/UAS-RyR RNAi*, (G-G') *OK-371-Gal4/ +; UAS-td-Tomato-GCaMP5G/ +*Dantrolene and (H-H') *OK371-Gal4, Asap$^{B52}$/OK371-Gal4, Asap$^{K23}$; UAS-td-Tomato-GCaMP5G/UAS-itpr RNAi*+Dantrolene expressing GCaMP5G (green) and td-Tomato (magenta) fluorescent protein. **(I-P)** Intensity plot profile for GCaMP5G (green) and tdTomato (magenta) across the bouton (shown in A'-H') as a thin line. The scale bar in H' for (A-H') represents 4 µm. **(Q)** Histogram showing the % of fluorescence ratio of GCaMP5G and td-Tomato per µm$^2$ bouton area in the indicated genotypes. ***$p$=0.0001. The statistical analysis was done using one-way ANOVA followed by post-hoc Tukey's multiple-comparison test. n=24–32 boutons per genotype. All values represent mean±SEM. The values for each quantification are shown in Table O in S1 Text. (TIF)

**S8 Fig. IP$_3$R and RyR heterozygous mutations partially suppress the *Asap* mutant phenotype. (A-F)** Representative traces of mEPSP and EPSP in (A) *w$^{1118}$* control, (B) *Asap$^{B52/K23}$*, (C) *itpr$^{90B.0}$/ +*, (D) *Asap$^{B52/K23}$; itpr $^{90B.0}$/ +*, (E) *RyR$^{E4340K}$/+* and (F) *Asap$^{B52}$, RyR$^{E4340K}$/Asap$^{K23}$* larvae. Scale bars for EPSPs (mEPSP) are x=200 ms (1000 ms) and y=10 mV (1 mV). **(G-I)** Histogram showing mEPSP amplitude (G), mEPSP frequency (H) and EPSP amplitude (I) from muscle 6 of A2 hemisegment in the indicated genotypes. ***$p$=0.0004 (mEPSP frequency), *$p$=0.03, **$p$=0.001 (EPSP amplitudes). The error bar represents the standard error of the mean (SEM). The statistical analysis was done using one-way ANOVA followed by post-hoc Tukey's test. ns, not significant. n=7–10 NMJ per genotype. All recordings included in the analysis have an input resistance greater than 5 MΩ. The values for each quantification are shown in Table P in S1 Text. (TIF)

**S9 Fig. Expression of Arf6$^{DN}$ or Arf6 RNAi in motor neurons rescues active zone number and cytoskeletal defects of *Asap* mutation. (A-F')** Representative confocal images of NMJ synapses at muscle 4 of A2 hemisegment showing the futsch loops in (A-A') *D42-Gal4/ +*control, (B-B') *Asap$^{K23/B52}$*, (C-C') *D42-Gal4/UAS-Arf6$^{DN}$*, (D-D') *Asap$^{K23/B52}$; D42-Gal4/ UAS-Arf6$^{DN}$*, (E-E') *UAS-Arf6 RNAi/ +; D42-Gal4/+* and (F-F') *UAS-Arf6 RNAi, Asap$^{B52/K23}$; D42-Gal4/ +*double immunolabeled with 22C10 (green) and a neuronal membrane marker, HRP (magenta). Scale bar in F' (for A-F') represents 4 µm. **(G)** Histogram showing the percentage of futsch positive loops from muscle 4 NMJ at A2 hemisegment in *D42-Gal4/ +*control, *Asap$^{K23/B52}$, D42-Gal4/UAS-Arf6$^{DN}$, Asap$^{K23/B52}$; D42-Gal4/UAS-Arf6$^{DN}$, UAS-Arf6$^{RNAi}$/ +; D42-Gal4/+* and *UAS-Arf6 RNAi, Asap$^{B52/K23}$; D42-Gal4/ +*animals. The error bar represents the standard error of the mean (SEM); the statistical analysis was done using one-way ANOVA followed by post-hoc Tukey's test. ***$p$=0.0001; ns, not significant. n=10–15 NMJ per genotype. All values represent mean±SEM. **(H-M')** Representative confocal images of NMJ synapses at muscle 4 of A2 hemisegment showing active zone density in (H-H') *D42-Gal4/ +*control, (I-I') *Asap$^{K23/B52}$*, (J-J') *D42-Gal4/UAS-Arf6$^{DN}$*, (K-K') *Asap$^{K23/B52}$; D42-Gal4/UAS-Arf6$^{DN}$*, (L-L') *UAS-Arf6$^{RNAi}$/ +; D42-Gal4/+* and (M-M') *UAS-Arf6 RNAi, Asap$^{B52/K23}$; D42-Gal4/ +*double immunolabeled with antibodies against active zone marker Brp (green) and GluRIII (magenta). The scale bar in M' (for H-M') represents 2.5 µm. **(N)** Histogram showing the number of Brp punctae per NMJ from muscle 4 NMJ at A2 hemisegment in *D42-Gal4/ +*control, *Asap$^{K23/B52}$, D42-Gal4/UAS-Arf6$^{DN}$, Asap$^{K23/B52}$; D42-Gal4/UAS-Arf6$^{DN}$, UAS-Arf6$^{RNAi}$/ +; D42-Gal4/+* and *UAS-Arf6 RNAi, Asap$^{B52/K23}$; D42-Gal4/ +*animals. The error bar represents the standard error of the mean (SEM); the statistical analysis was done using one-way ANOVA followed by post-hoc Tukey's test. ***$p$<0.0001; ns, not significant. n=6–13 NMJ per genotype. All values represent mean±SEM. **(O-T)** Representative confocal images of NMJ synapses at muscle 4 of A2 hemisegment showing Rab3 levels in (O) *D42-Gal4/ +*control, (P) *Asap$^{K23/B52}$*, (Q) *D42-Gal4/UAS-Arf6$^{DN}$*, (R) *Asap$^{K23/B52}$; D42-Gal4/UAS-Arf6$^{DN}$*, (S) *UAS-Arf6 RNAi/ +; D42-Gal4/+* and (T)

*UAS-Arf6 RNAi, Asap^B52/K23; D42-Gal4/* +double immunolabeled with antibodies against Rab3 (magenta) and HRP (green). The scale bar in T represents 2.5 μm. **(U)** Histogram showing the Rab3 levels from muscle 4 NMJ at A2 hemisegment in *D42-Gal4/* +control, *Asap^K23/B52*, *D42-Gal4/UAS-Arf6^DN*, *Asap^K23/B52; D42-Gal4/UAS-Arf6^DN*, *UAS-Arf6^RNAi/* +*; D42-Gal4/+* and *UAS-Arf6 RNAi, Asap^B52/K23; D42-Gal4/* +animals. The error bar represents the standard error of the mean (SEM); the statistical analysis was done using one-way ANOVA followed by post-hoc Tukey's test. ***$p = 0.0001$; ns, not significant. $n = 28–32$ boutons per genotype. All values represent mean ± SEM. The values for each quantification are shown in Table Q in S1 Text.
(TIF)

**S10 Fig. Neuronal loss of IP$_3$ signaling components enhances Rab3-dependent compensation of active zone stability in *Asap* mutants. (A-H')** Confocal images of NMJ synapses at muscle 4 of A2 hemisegment in *D42-Gal4/+* (A-A'), *Asap^B52/K23* (B-B'), *D42-Gal4/UAS-plc-β RNAi* (C-C'), *Asap^B52/K23; D42-Gal4/UAS-plc-β RNAi* (D-D'), *D42-Gal4/UAS-itpr* RNAi (E-E'), *Asap^B52/K23; D42-Gal4/UAS-itpr RNAi* (F-F'), *D42-Gal4/UAS-RyR* RNAi (G-G') and *Asap^B52/K23; D42-Gal4/UAS-RyR RNAi* (H-H') double immunolabeled with antibodies against active zones marker Bruchpilot, Brp (green) and HRP (magenta). The scale bar in H' represents 2.5 μm for A-H'. **(I)** Histogram showing the levels of Brp per μm$^2$ area of bouton from muscle 4 at the A2 hemisegment in the indicated genotypes. ***$p = 0.001$, ***$p = 0.0001$. $n = 26–30$ boutons per genotype. All values represent mean ± SEM. **(J)** Histogram showing the number of Brp punctae per μm$^2$ area of bouton from muscle 4 NMJ at A2 hemisegment in the indicated genotypes. **$p = 0.01$, $n = 26–30$ boutons per genotype. All values represent mean ± SEM. **(K-R)** Confocal images of NMJ synapses at muscle 4 of A2 hemisegment in *D42-Gal4/+* (K), *Asap^B52/K23* (L), *D42-Gal4/UAS-plc-β RNAi* (M), *Asap^B52/K23; D42-Gal4/UAS-plc-β RNAi* (N), *D42-Gal4/UAS-itpr* RNAi (O), *Asap^B52/K23; D42-Gal4/UAS-itpr RNAi* (P), *D42-Gal4/UAS-RyR* RNAi (Q) and *Asap^B52/K23; D42-Gal4/UAS-RyR RNAi* (R) double immunolabeled with antibodies against Rab3 (magenta) and HRP (green). The scale bar in R represents 2.5 μm for K-R. **(S)** Histogram showing the levels of Rab3 per μm$^2$ area of bouton from muscle 4 NMJ at A2 hemisegment in the indicated genotypes. ***$p = 0.001$. $n = 25–31$ boutons per genotype. All values represent mean ± SEM. The values for each quantification are shown in Table R in S1 Text.
(TIF)

**S11 Fig. Validation of RNAi efficiency by qPCR reduced transcript levels of Arf6, *Itpr*, *RyR*, *Plc21C* and *Letm1* transcripts in respective RNAi lines.** Quantitative RT-PCR showing transcript levels of *Arf6, Plc21C*, *Letm1*, *Itpr*, and *RyR* in *actin5C-Gal4*-driven corresponding RNAi lines, as indicated in the figure. The RNAi lines for *Arf6, Plc21C*, and *RyR* showed a ~50% reduction in transcript levels compared to the control animals. For *Itpr* and *Letm1*, a ~40% reduction in transcript levels was observed compared to the *actin5C-Gal4* control (*actin5C-Gal4/+)*. All values represent mean ± SEM. **$p = 0.01$, ***$p = 0.001$. The values for each quantification are shown in Table S in S1 Text.
(TIF)

**S1 Text. Table A.** The table shows the analysis of total bouton number, average bouton area, average bouton size, average inter-bouton diameter, and Futsch loops on muscle 4 of A2 hemisegment in various genetic combinations as indicated. Values are represented as mean±s.e.m. The control mean is compared with the mutant, while the mutant mean is compared with various rescue combinations. These values relate to Fig 1. **Table B.** The table shows values of mEPSP amplitude, mEPSP frequency, EPSP amplitude, Quantal content, the average number of Brp punctae per NMJ, GluRIII cluster area, number of synaptic failure events, Brp intensity/ μm$^2$ area of bouton, CAC intensity/ μm$^2$ area of bouton, and CAC density μm$^2$ area of bouton in the indicated genotypes. Values are represented as mean±s.e.m. The control mean is compared with the mutant, while the mutant mean is compared with the rescue combination. These values relate to Fig 2.
(DOCX)

## Acknowledgments

We acknowledge the Developmental Studies Hybridoma Bank (Iowa, USA) for antibodies used in this study and the Bloomington *Drosophila* Stock Center for fly stocks. We thank Drs. Ruth Johnson for Asap antibodies, Aaron DiAntonio for Rab3 and GluRIII antibodies and Tonny Harris for UAS-Asap-mcherry lines. We thank Drs. Masayuki Koganezawa and Kartik Venkatachalam for providing us with UAS-IP3-Sponge.m30 and tdTomato-P2A-GCaMP5G lines. We thank P Srikanth for qPCR data and help with image processing. We thank Prof. Gaiti Hasan, Sunando Datta, and the VK and Frank lab members for their helpful comments and discussions on this study.

## Author contributions

**Conceptualization:** C Andrew Frank, Vimlesh Kumar.

**Data curation:** Bhagaban Mallik, Vimlesh Kumar.

**Formal analysis:** Bhagaban Mallik, C Andrew Frank, Vimlesh Kumar.

**Funding acquisition:** C Andrew Frank, Vimlesh Kumar.

**Investigation:** Bhagaban Mallik, Shikha Kushwaha, Anjali Bisht, Harsha MJ.

**Methodology:** Bhagaban Mallik.

**Project administration:** C Andrew Frank, Vimlesh Kumar.

**Supervision:** C Andrew Frank, Vimlesh Kumar.

**Validation:** Bhagaban Mallik, Anjali Bisht.

**Visualization:** Bhagaban Mallik, Shikha Kushwaha, Anjali Bisht, Harsha MJ.

**Writing – original draft:** Bhagaban Mallik, Vimlesh Kumar.

**Writing – review & editing:** Bhagaban Mallik, Harsha MJ, C Andrew Frank, Vimlesh Kumar.

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
