## [Decision Letter · Decision Letter 0]

1 Jun 2025

PGENETICS-D-25-00364

An ArfGAP-Dependent Signaling Modulates Synaptic Plasticity via IP3-Regulated Calcium Release from the Endoplasmic Reticulum

PLOS Genetics

Dear Dr. Kumar,

Thank you for submitting your manuscript to PLOS Genetics. After careful consideration, we feel that it has merit but does not fully meet PLOS Genetics's publication criteria as it currently stands. Therefore, we invite you to submit a revised version of the manuscript that addresses the points raised during the review process.

Please submit your revised manuscript within 60 days Jul 31 2025 11:59PM. If you will need more time than this to complete your revisions, please reply to this message or contact the journal office at plosgenetics@plos.org. Please include the following items when submitting your revised manuscript:

We look forward to receiving your revised manuscript.

Kind regards,

Guang-Chao Chen

Academic Editor

PLOS Genetics

Fengwei Yu

Section Editor

PLOS Genetics

Aimée Dudley

Editor-in-Chief

PLOS Genetics

Anne Goriely

Editor-in-Chief

PLOS Genetics

**Additional Editor Comments :**

The manuscript has been reviewed by four experts in the field. While they all find the study interesting, they have requested additional experimental data to support the conclusions. Please consider their comments carefully and address all the points raised.

**Journal Requirements:**

1) Please provide an Author Summary. This should appear in your manuscript between the Abstract (if applicable) and the Introduction, and should be 150-200 words long. The aim should be to make your findings accessible to a wide audience that includes both scientists and non-scientists. Sample summaries can be found on our website under Submission Guidelines:

https://journals.plos.org/plosgenetics/s/submission-guidelines#loc-parts-of-a-submission

- TM on page: 19.

Potential Copyright Issues:

i) Figures 3A, 4F, 4M, 4N, 5A, and 8. Please confirm whether you drew the images / clip-art within the figure panels by hand. If you did not draw the images, please provide (a) a link to the source of the images or icons and their license / terms of use; or (b) written permission from the copyright holder to publish the images or icons under our CC BY 4.0 license. Alternatively, you may replace the images with open source alternatives. See these open source resources you may use to replace images / clip-art:

2) If any authors received a salary from any of your funders, please state which authors and which funders.

6) Please ensure that the funders and grant numbers match between the Financial Disclosure field and the Funding Information tab in your submission form. Note that the funders must be provided in the same order in both places as well. Currently, the order of the grants is different in both places.

**Reviewers' comments:**

Reviewer's Responses to Questions

Reviewer #1: IP3 and ryanodine receptors are known to regulate synaptic strength. However, the upstream mechanisms that regulate IP3 and ryanodine receptors are not well understood. Malik et al. investigated the role of Drosophila Asap in synapse development and calcium homeostasis. The found that loss of Asap resulted in structural changes (increased bouton size, reduced bouton number, and higher active zone density), increased mEJP frequency, reduced synaptic failures under low extracellular Ca2+, and elevated basal presynaptic calcium levels. Genetic epistatic experiments suggest that Asap regulates Arf6 and restricts calcium release through the IP3-dependent pathway. Overall, this is an interesting study that provides mechanistic insight into upstream mechanisms of IP3 and ryanodine receptors that affect synaptic structure and function. Below are several points that I think would improve the manuscript.

Major points

1. Supplemental Figure 2 indicates that the Asap antibodies are not specific for Asap protein, given that staining is observed in the Asap null (Fig. S2C). The fact that pre-absorption of antibody with pure Asap protein eliminated all staining doesn’t mean that any of the staining was Asap protein. In Fig 1G, the authors performed Western blots on fillet preps and could detect Asap protein in the mef2-GAL4 rescue line. If the authors ran a Western on control larval fillet preps and were also able to detect Asap protein this would suggest that Asap is likely expressed at the nmj.

2. How was GluRII cluster area calculated?

3. Increased basal presynaptic calcium levels likely explain the increased mEJP frequency seen in Asap mutants. Did the authors look at presynaptic calcium levels response to stimulation to determine if changes in presynaptic calcium were responsible for the increased EJP amplitudes. This is important in linking changes in evoked neurotransmission to changes in presynaptic calcium. This would be a better experiment than doing electrophysiology in the presence of BAPTA-AM.

4. What concentration of BAPTA-AM was used in Fig. 4? It’s surprising that BAPTA-AM did not reduce evoked neurotransmission in controls.

5. It would be good to confirm that UAS and GAL4 controls do not rescue Asap phenotypes in Fig 4, 5, 6 and 7. There could be insertional effects or leaky expression from UAS lines.

6. Also, have Ryanodine, Letm1, PLCB, and Arf6 RNAi lines been verified to show that they do knock down Ryanodine, Letm1, PLCB, and Arf6 respectively.

Reviewer #2: This manuscript uncovers a potentially interesting role of ArfGAP in synaptic development and neurotransmission. While the authors provide a substantial amount of data, several critical issues need to be addressed to strengthen the conclusions:

1. The genotypes of the control groups in each figure are not clearly indicated. This information is essential for interpreting the experimental results.

2. There are notable inconsistencies in mEPSP, EPSP amplitudes, and quantal content (QC) across Figures 1, 4, 5, and 7. Are these differences due to varying extracellular Ca²⁺ concentrations or genetic backgrounds? Such methodological details must be clearly stated.

3. Concerns regarding Figure 4

- The identities and descriptions of panels 4O and 4P are missing from the figure legend and main text.

- The data suggest that loss of aspa increases resting Ca²⁺ levels, a slow calcium dynamic. In this context, BAPTA-AM, which preferentially chelates fast calcium transients, may not be appropriate. EDTA-AM would be more suitable for buffering slow Ca²⁺ changes.

- BAPTA-AM reduces EPSP amplitude in the aspa mutant; however, it is unclear why a similar effect is not observed in the control.

- Since RNAi knockdown of IP3R and RyR already reduces mEPSP frequency in the control, it is unclear whether these ER calcium channels are solely responsible for the elevated mEPSP frequency in aspa mutants. Testing these effects in heterozygous Itpr and RyR mutant backgrounds would help clarify this.

- Does knockdown of IP3R or RyR also normalize resting Ca²⁺ levels in the aspa mutant? This is important data.

- A double knockdown condition (2xD42>IP3R-RNAi + RyR-RNAi) should be included.

- The effect of Letm1 knockdown in the control background is missing, making it difficult to interpret its role.

4. Concerns regarding Figure 5

- Knockdown of plcβ reduces mEPSP frequency similarly in both control and aspa mutants, casting doubt on its proposed role in mediating the aspa-dependent increase in miniature events. Furthermore, no data on resting Ca²⁺ levels are provided in this figure.

5. Concerns regarding Figure 6

- The effect of Arf-RNAi, Arf6-DN, or Arf6-CA expression alone on bouton morphology is not shown. Without these controls, it is difficult to conclude whether increased Arf activity is causally related to the bouton phenotype in aspa mutants.

6. Concerns regarding Figure 7

- As in Figure 6, the effects of Arf6-RNAi expression alone should be presented.

- It is unclear whether Arf6-DN expression also reduces the elevated resting Ca²⁺ levels in aspa mutants.

7. It is not investigated whether active zone intensity and VGCC activity may contribute to the phenotype of aspa mutant. Since aspa mutants show increased active zone intensity, it is possible that this change enhances VGCC activity and contributes to the elevated resting Ca²⁺ concentration. The authors should test whether knockdown of IP3R, RyR, Arf6, or Arf6-DN also rescues this active zone phenotype.

8. It is unclear the potential role of microtubules in ER channel regulation. Given the established link between ER organization and microtubule dynamics, it is plausible that altered microtubule architecture underlies the observed increase in ER Ca²⁺ channel activity. This possibility should be explored.

9. The authors claim that actin and moesin structures are unaltered in aspa mutants (Supplemental Figure S3), yet only acetylated tubulin staining is shown. This statement should be revised accordingly.

10. The abstract refers to “pharmacological and genetic manipulations,” yet no pharmacological experiments are included. This statement should be corrected.

11. The error bars in Figure 3 appear to represent standard deviation but are labeled as SEM. This should be corrected for consistency and clarity.

Reviewer #3: The authors examine the role of Arf-specific GAP Asap in modulating IP3-regulated calcium release from the ER at the Drosophila NMJ. They generated dAsap mutants and other genetic reagents to perform phenotypic and quantitative analyses in different morphological and electrophysiological assays. The authors show loss of dAsap elevates resting calcium in synaptic terminals, enhances vesicle release, and disrupts both synaptic morphology and microtubule organization. Further genetic interaction experiments with PLCβ and dArf6 suggest a novel mechanism where dAsap inhibits Arf6-PLCβ-IP3 signaling via its ArfGAP activity, thereby controlling calcium release from ER.

The work presented is interesting and I particularly enjoy the extensiveness of the assays the authors employed. Overall, the phenotypic characterization of dAsap mutants using morphological markers and electrophysiological assays is quite extensive. However, the evidence of functional link between dAsap and dArf6 or PLCβ is rather weak. In Fig.5 F-I, PLCβ RNAi itself is not statistically different from Control, raising the possibility of a compensatory effect of PLCβ RNAi upon loss of dAsap. Moreover, in Fig.6-7 and Fig.S4-5, the experiments were done for dArf6 DN in Asap mutant background, but no data for dArf6 mutants or dArf6 DN alone. Therefore, the authors' conclusion regarding the dAsap-dArf6 regulatory relationship extends beyond what the current data can substantiate.

Another criticism is the lack of proper genetic controls throughout the manuscript. The authors used the Gal4/UAS system extensively but seemed not to consider the controls for these manipulations. For instance, in the rescue experiments at least UAS-dAsap alone, i.e. without the Gal4 drivers, should be performed as a proper control. For RNAi or overexpression of dArf6-DN, Gal4 alone and UAS-RNAi or UAS-dArf6-DN should be performed.

Minor point:

The authors attempted to perform epistatic analysis of dArf1 and dAsap in a dAsap mutant background by overexpressing dArf1-DN, but the animals did not survive to the 3rd instar larval stage. This situation can be circumvented by using the classical MARCM method. MARCM can generate dAsap mutant clones and simultaneously overexpress dArf1-DN and mCD8GFP in a dAsap-/+ mosaic background. The desired clones can be identified by GFP+ expression.

Reviewer #4: The manuscript by Mallik et al addresses the role of Asap and Arf6 in modulating IP3-regulated calcium release at the larval NMJ. The authors generate Asap mutants and report morphological and electrophysiological phenotypes, including larger boutons with increased number of active zones, increased mini frequency and increased evoked responses. They correlate these defects with elevated levels of cytoplasmic calcium at resting state, and confirmed this by direct measurements with a GCaMP sensor. The authors use genetics, pharmacology and electrophysiology to work through the relevant pathways downstream Asap. They report that Asap limits RyR, IP3 and plc-beta activities, which together control the release of calcium from ER, but not from mitochondria. Finaly, the authors compare the Asap/IP3 signaling phenotypes with combination of Asap/Arf6 l-o-f (Arf6 RNAi or a dominant negative transgene in the Asap mutant background). These epistasis experiments indicate that Asap l-o-f increases the intracellular Ca2+ in an Arf6-dependent manner; also, the NMJ phenotypes observed in Asap mutants are Arf6-dependent.

The findings that Asap/Arf6/IP3 signaling control calcium release and thus regulate the NMJ function are novel and interesting. The genetics and epistasis arguments usually provide a powerful means to sort through various pathway components. But the authors left out several experiments and, consequently, the reader is presented with correlative data rather than direct epistasis results. For example, the direct connection between Arf6 and IP3 is inferred but never tested directly. There are also a number of incorrect statements and panels that must be revised before this manuscript could be considered for publication.

Major issues:

1) The epistasis experiments are incomplete.

The authors showed two panels with Arf6-CA overexpression in Asap mutants (Figure 6C) but they do not mention these results in the manuscript. This Arf6-CA transgene must be overexpressed by itself, in a wild-type background, and the NMJ morphology and function of these animals (i.e. D42> Arf6-CA) must be compared to that of Asap l-o-f.

2) The Arf6-CA transgene should also be combined with IP3-sponge, Ryr-RNAi and plc-beta RNAi to directly test the epistatic relationships Arf6 and the other pathway components.

As it stands now, the authors show no direct connection between Arf6 and IP3, etc. The epistasis must be included here to support the model proposed.

3) The authors don't provide any explanation for the increased Brp puncta. This is unfortunate and should be addressed.

For example, a paper that the authors cite, Pelletan et al, 2015, showed that active ARF6 increases the exchange of GDP for GTP on Rab3A. In addition, it was previously demonstrated that Rab3 (the fly ortholog of vertebrate Rab3A) dynamically controls the composition of the presynaptic release machinery at larval NMJ, including the Brp levels (PMID:20005823). This may explain why Asap l-o-f (and therefore Arf6 g-o-f) mutants have increased number of Brp puncta.

A somewhat satisfying experiment required here would be to monitor the Rab3 levels in all the genetic manipulations aimed at examining Brp puncta. The published antibody is available upon request.

4) On the same topic, ideally, the authors should include a simple set of experiments confirming (or infirming) a direct connection between Arf6 and Rab3 in causing the Brp phenotypes reported here. These results will tremendously strengthen the authors' arguments and also take care of a line of observations left unexplained.

5) Fig. 3C-J/ Table S3

The statement "Presynaptic homeostatic plasticity remains intact in Asap mutant" is incorrect and should be revised.

Based on the values provided in Table S3, the Asap l-o-f larvae have a 1.34-fold increase in QC in response to PhTx. This is less than the required 1.5-fold, which is the classic threshold used to mark the expression of PHP. Thus, the Asap mutants are in fact PHP deficient. This fact provides an even stronger argument to search for alternative explanations for the observed increased basal neurotransmission. The authors must correct this paragraph. Please report QC ratios for clearer results.

Other concerns:

1) Fig. 1A- please show the gene diagram to scale; include a scale bar.

Also, please add S2A as an additional panel in the main Fig. 1. This will facilitate readers' understanding of the narrative related to Asap domains and the antigen used for antibody generation.

2) The Brp puncta appear similar between Asap mutants and wt control in Fig. 2I, but they are much more intense in the Asap mutants in Fig. 3M. This is concerning.

Also, in both Figs 2I and 3M, the authors must report the intensity of Brp puncta not only the puncta number per NMJ.

- Please change panel 2I with one from a specimen that is not stretched. As is, this panel is confusing.

- Please add an additional Brp-Fire-LUT panel in Fig 2H-J; this will capture the Brp intensities.

3) In Fig. 2M, the authors plot GluRIII cluster area, even though they probably measured the cluster diameter. (The Materials and Methods lack any description for this metrics.) Please report the GluRIII cluster diameter - it doesn't help to exacerbate the difference.

4) Show muscle resistance for all the electrophysiology recordings in the supplementary tables. Also, indicate the extracellular Ca2+ concentrations.

5) The synapse diagrams are colored in blue/white (Fig. 3), blue/blue (Fig. 4), white (Fig. 5A) or pink/pink (Fig. 8). This is neither appropriate nor useful. Please keep things consistent. Also, many of these diagrams do not need the muscle side. Please simplify.

6) Fig. 3A and the associated main text introduce GluRIIA as the only receptor affected by PhTx. This is incorrect (please see PMID: 39602131). Please revise and also simplify the diagram accordingly.

7) Please revise the traces and data shown in Figure S4. They are not on the same scale.

The spacing between pulses (in panels C and E) varies. Also, the amplitudes of traces shown in panel G do not match the data plotted in panel H. For example, the last trace has an amplitude >40 mV, but there is no datapoint in panel H with such a large amplitude value for that genotype.

8) The number of Brp puncta varies 2-fold between the Asap l-o-f larvae in Fig. 2K and the same genotype revisited in Fig. S5 panel J. This is concerning, please explain.

9) Please rephrase the sentence below for clarity.

"The multidomain containing ArfGAP with SH3 domain, Ankyrin repeat and BAR domain Protein (ASAP) subfamily proteins, which contain a BAR and an ArfGAP domain, hydrolyze Arf6.GTP through its GTPase-activating activity in vitro (Ismail et al., 2010; Kushwaha et al., 2024)."

10) To troubleshoot the anti-Asap high background staining, the authors should try to pre-adsorb their sera with Asap l-o-f larvae.

11) pg. 12- Remove "from ER" from the sentence below - as this was not addressed yet up to this point.

"Together, these results are consistent with the idea that elevated intracellular calcium release from ER and increased active zone numbers in Asap mutants contribute to maintaining robust neurotransmission."

**Have all data underlying the figures and results presented in the manuscript been provided?**

Reviewer #1: Yes

Reviewer #2: Yes

Reviewer #3: Yes

Reviewer #4: Yes

PLOS authors have the option to publish the peer review history of their article (what does this mean? ). If published, this will include your full peer review and any attached files.

**Do you want your identity to be public for this peer review?** For information about this choice, including consent withdrawal, please see our Privacy Policy .

Reviewer #1: No

Reviewer #2: **Yes:** Chi-Kuang Yao

Reviewer #3: No

Reviewer #4: No

**Figure resubmission:**
---

## [Decision Letter · Decision Letter 1]

18 Dec 2025

PGENETICS-D-25-00364R1

An ArfGAP-Dependent Signaling Modulates Synaptic Plasticity via IP3-Regulated Calcium Release from the Endoplasmic Reticulum

PLOS Genetics

Dear Dr. Kumar,

Thank you for submitting your manuscript to PLOS Genetics. After careful consideration, we feel that it has merit but does not fully meet PLOS Genetics's publication criteria as it currently stands. Therefore, we invite you to submit a revised version of the manuscript that addresses the points raised during the review process.

Please submit your revised manuscript within by Jan 17 2026 11:59PM. If you will need more time than this to complete your revisions, please reply to this message or contact the journal office at plosgenetics@plos.org. Please include the following items when submitting your revised manuscript:

We look forward to receiving your revised manuscript.

Kind regards,

Guang-Chao Chen

Academic Editor

PLOS Genetics

Fengwei Yu

Section Editor

PLOS Genetics

Aimée Dudley

Editor-in-Chief

PLOS Genetics

Anne Goriely

Editor-in-Chief

PLOS Genetics

**Additional Editor Comments:**

Reviewer 1 and Reviewer 4 still have some minor comments; please address them accordingly.

**Journal Requirements:**

**Reviewers' comments:**

Reviewer's Responses to Questions

**Comments to the Authors:**

Reviewer #1: Overall, the revised manuscript has been improved and the authors have addressed the majority of my comments. I do think the manuscript would have been further strengthened by looking at presynaptic calcium levels in response to stimulation to determine if changes in presynaptic calcium were responsible for the increased EJP amplitudes

Reviewer #2: The authors have satisfactorily addressed my previous concerns. The revised manuscript is suitable for publication.

Reviewer #3: I commend the authors on their thorough and balanced responses to my concerns, and now feel that the manuscript is appropriate for publication.

Reviewer #4: Congratulations on a solid piece of work. The revised manuscript is much improved.

However, the original EPSP trace in Fig 3E (but not the mEPSP) was replaced with a different trace, the amplitude of which is lower than any data point presented in the accompanying the bar graph (3I). Please correct.

**Have all data underlying the figures and results presented in the manuscript been provided?**

Reviewer #1: Yes

Reviewer #2: Yes

Reviewer #3: Yes

Reviewer #4: None

PLOS authors have the option to publish the peer review history of their article (what does this mean? ). If published, this will include your full peer review and any attached files.

**Do you want your identity to be public for this peer review?** For information about this choice, including consent withdrawal, please see our Privacy Policy .

Reviewer #1: No

Reviewer #2: **Yes:** Chi-Kuang Yao

Reviewer #3: No

Reviewer #4: No

**Figure resubmission:**
---

## [Editor Report · Decision Letter 2]

13 Jan 2026

Dear Dr Kumar,

We are pleased to inform you that your manuscript entitled "An ArfGAP-Dependent Signaling Modulates Synaptic Plasticity via IP3-Regulated Calcium Release from the Endoplasmic Reticulum" has been editorially accepted for publication in PLOS Genetics. Congratulations!

Yours sincerely,

Guang-Chao Chen

Academic Editor

PLOS Genetics

Fengwei Yu

Section Editor

PLOS Genetics

Aimée Dudley

Editor-in-Chief

PLOS Genetics

Anne Goriely

Editor-in-Chief

PLOS Genetics

BlueSky: @plos.bsky.social

Comments from the reviewers (if applicable):

**Data Deposition**

http://datadryad.org/submit?journalID=pgenetics&manu=PGENETICS-D-25-00364R2

**Press Queries**

---

## [Editor Report · Acceptance letter]

PGENETICS-D-25-00364R2

An ArfGAP-Dependent Signaling Modulates Synaptic Plasticity via IP3-Regulated Calcium Release from the Endoplasmic Reticulum

Dear Dr Kumar,

We are pleased to inform you that your manuscript entitled "An ArfGAP-Dependent Signaling Modulates Synaptic Plasticity via IP3-Regulated Calcium Release from the Endoplasmic Reticulum" has been formally accepted for publication in PLOS Genetics! Your manuscript is now with our production department and you will be notified of the publication date in due course.

With kind regards,

Zsofia Freund

PLOS Genetics

On behalf of:
